



**Global meta-analysis of the relationship between soil organic matter and crop yields**
Emily E. Oldfield[1*], Mark A. Bradford[1], Stephen A. Wood[1,2]
[1]School of Forestry and Environmental Studies, Yale University, 370 Prospect Street,
New Haven, CT 06511, USA
[2]The Nature Conservancy, Arlington, VA 22201, USA
Correspondence: Emily E. Oldfield, School of Forestry and Environmental Studies, Yale
University, 370 Prospect St, New Haven, CT, 06511, USA
Email: emily.oldfield@yale.edu
Co-author emails: MAB (mark.bradford@yale.edu); SAW (stephen.wood@tnc.org)
Keywords: crop productivity, soil carbon, soil organic matter, sustainable intensification,
yield gap



**Abstract**
Resilient, productive soils are necessary to sustainably intensify agriculture to increase
yields while minimizing environmental harm. To conserve and regenerate productive
soils, the need to build and maintain soil organic matter (SOM) has received considerable
attention. Although SOM is considered key to soil health, its relationship with yield is
contested because of local-scale differences in soils, climate, and farming systems. There
is a need to quantify this relationship to set a general framework for how soil
management could potentially contribute to the goals of sustainable intensification. We
developed a quantitative model exploring how SOM relates to crop yield potential of
maize and wheat in light of co-varying factors of management, soil type, and climate. We
found that yields of these two crops are on average greater with higher concentrations of
SOC. However, yield increases level off at ~2% SOC. Nevertheless, approximately two
thirds of the world's cultivated maize and wheat lands currently have SOC contents of
less than 2%. Using this regression relationship developed from published empirical data,
we then estimated how an increase in SOC concentrations up to regionally-specific
targets could potentially help reduce reliance on nitrogen (N) fertilizer and help close
global yield gaps. Potential N fertilizer reductions associated with increasing SOC
amount to 7% and 5% of global N fertilizer inputs across maize and wheat fields,
respectively. Potential yield increases of 10±11% (mean±SD) for maize and 23±37% for
wheat amount to 32% of the projected yield gap for maize and 60% of that for wheat. Our
analysis provides a global-level prediction for relating SOC to crop yields. Further work
employing similar approaches to regional and local data, coupled with experimental work

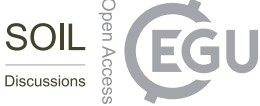

to disentangle causative effects of SOC on yield and vice-versa, are needed to provide
practical prescriptions to incentivize soil management for sustainable intensification.



## 1 Introduction

The pressure to increase crop production has resulted in the expansion of land area

dedicated to agriculture and the intensification of cropland management through practices

such as irrigation and fertilization. These practices have led to degradation of land and

waters prompting "sustainable intensification" initiatives to increase yields on existing

farmland while decreasing the environmental impact of agriculture (Foley et al., 2011;

Godfray et al., 2010; Mueller et al., 2012). One sign of land degradation is the loss of soil

organic matter (SOM) (Reeves, 1997). Re-building SOM in agricultural lands holds the

promise of improving soil fertility, as SOM affects many properties of soils, including

their ability to retain water and nutrients, to provide structure promoting efficient

drainage and aeration, and to minimize loss of top-soil via erosion (Reeves et al., 1997;

Robertson et al., 2014). As such, managing SOM to ensure stable and long-lasting crop

productivity, and to decrease reliance on external inputs such as mineral fertilizers and

irrigation, has been identified as a critical component of sustainable intensification (Foley

et al., 2011). Yet the emphasis on soil management has remained qualitative, meaning

that the potential contribution of building SOM as a means to increase crop production

and minimize the environmental impact of agriculture has not yet been broadly quantified

(Adhikari and Hartemink, 2016; Chabbi et al., 2017; Hatfield et al., 2017).

A primary hurdle to managing SOM for sustainable intensification is the lack of

predictive, quantitative targets of SOM for specific agricultural and environmental

objectives (Herrick, 2000; NRC, 2010). While several studies show correlations between

SOM and yield (Culman et al., 2013; de Moraes Sa et al., 2014; Lucas and Weil, 2012;

Stine and Weil, 2002), it remains unclear how much yield could be expected to increase



per unit change in organic matter (Herrick, 2000; NRC, 2010). Establishing these
quantitative metrics is challenging because research shows increases (Bauer and Black,
1992), decreases (Bhardwaj et al., 2011), and no change (Hijbeek et al., 2017) in yields
with increased SOM. This lack of a general relationship is likely the result of a number of
interacting factors related to management, climate, and soil type that can confound the
SOM-yield relationship. This confusion has led some to claim that the amount of SOM is
unnecessary for crop yields, so long as there is sufficient N fertilizer (Hijbeek et al.,
2017; Loveland and Webb, 2003; Oelofse et al., 2015); whereas others highlight the need
to build SOM to increase crop yields while minimizing environmental harm (Lal, 2004).
The growing momentum to launch global scale initiatives to manage SOM (Banwart et
al., 2014; Lal, 2004; Minasny et al., 2017; Zomer et al., 2017) suggests the need to test
competing claims about the effects of SOM on these agricultural and environmental
outcomes.

One could critique the effort to establish a global-level understanding of the

SOM-yield relationship on the grounds that farm-level responses are necessarily
heterogeneous and poorly predicted by global assessments. Yet, global initiatives for
managing SOM could create policy environments that stimulate regional- and local-
prescriptions for SOM levels that inform practice (Chabbi et al., 2017; Minasny et al.,
2017; Zomer et al., 2017). Whereas it is difficult to disentangle the extent to which SOM-
yield relationships are driven by SOM effects on yield, as opposed to yield (i.e. higher
plant carbon inputs) effects on SOM, there is nevertheless experimental evidence
showing that building SOM positively affects yield (Bauer and Black, 1994; Majumder et
al., 2008; Oldfield et al., 2017). In addition, numerous soil properties that relate to soil

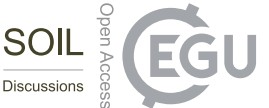

fertility, such as water holding capacity, respond positively to increasing SOM and in
turn are expected to increase yields (Williams et al., 2016). As such, correlative SOM-
yield relationships suggest the potential – but likely not the true – effect of SOM on yield.

We developed a quantitative model exploring how SOM relates to crop yield

potential in light of co-varying factors of management, soil type, and climate. The aim is
that this model can then be used to inform actionable and evaluable targets for soil
management as a central component of sustainable intensification efforts. We quantified
the relationship between SOM (measured as SOC, a common proxy for SOM) and yield
at a global level using data from published studies (Fig. 1). We focused our analyses on
wheat and maize, two common staple crops that (along with rice) constitute two-thirds of
the energy in human diets (Cassman, 1999). Along with SOC, we modeled the effects on
crop yields of several factors widely reported in yield studies: N input rate, irrigation, pH,
soil texture (% clay), aridity, crop type (i.e. wheat or maize), and latitude (as a proxy for
growing-season day length). The data informing our model came from empirical studies
that capture local scale variation in these variables, and hence we interpret our results in
light of the correlative nature of the database we assembled. Using the resulting multiple-
regression relationship, we then estimated how an increase in SOC concentrations up to
regionally-specific target thresholds might affect global yields. Our overarching aim was
then to estimate the potential extent to which restoring SOC in global agricultural lands
could help close global yield gaps and potentially help reduce reliance on – and the
negative effects of – N fertilizer.

**2 Results and Discussion**



## 2.1 The relationship between SOC and yield

At the global level and focusing specifically on the potential effect size of SOC on yield,

we found that the largest gains in yield occur between SOC concentrations of 0.1 to

2.0%. For instance, yields are 1.2 times higher at 1.0% SOC than 0.5% SOC (Fig. 2).

Gains in yield leveled off (i.e. the slope between yield and SOC is <0.25, when

controlling for N input) at a concentration of approximately 2% SOC (Fig. 2). Two

percent SOC has previously been suggested as a critical threshold, with values below this

concentration threatening the structure, and ultimately, the ability of a soil to function

(Kemper and Koch, 1966). Importantly, the asymptotic relationship between SOC and

yield lends support to the idea that building SOC will increase yields – at least to a

certain extent – as opposed to simply being an outcome of higher yields. That is, higher

yields might be expected to lead to greater plant carbon inputs to soils, but given that 2%

SOC is well below the carbon saturation point for most soils (Castellano et al., 2015), the

asymptote supports a causal effect of SOC.

It has been suggested that there is no evidence for 2% SOC being a critical

threshold for productivity, as long as there is sufficient mineral fertilizer to support crop

production (Edmeades, 2003; Loveland and Webb, 2003; Oelofse et al., 2015). Such

conclusions deem the amount of SOM as substitutable by mineral fertilizers for crop

growth, but are inconsistent with the motivation for sustainable intensification to

minimize environmental harm caused by mineral fertilizers in relation to emissions of

greenhouse gases and eutrophication of waters (Vitousek et al., 2009). This logic also

does not account for the other co-benefits associated with building SOM in agricultural

lands such as reductions in nutrient run-off, drought resistance, and yield stability



(Robertson et al., 2014). Specific field and regional studies have shown a similar pattern
as that observed from our global analysis: there exists a positive relationship between
SOC and yield that starts to level off at ~2% SOC (Kravchenko and Bullock, 2000; Pan
et al., 2009; Zvomuya et al., 2008). Our analysis suggests that this relationship holds on
average at the global scale and when N fertilization is controlled for.

Ninety-one percent of the published studies used for our analysis were carried out

in fields with less than 2% SOC, with a mean of 1.1%. To see whether these observations
in SOC distribution reflected global patterns, we used globally gridded data on crop yield
and SOC (to a depth of 15 cm) (Hengl et al., 2014; Monfreda et al., 2008). We found that,
by both area and production, two thirds of maize and wheat cultivation takes place on
soils with less than 2% SOC (Fig. 3). Indeed, a recent analysis estimates that agricultural
land uses (including cropland and grazing) have resulted in a loss of 133 Pg of carbon
over the past 12,000 years of human land use (Sanderman et al., 2017). There appears to
be, therefore, significant opportunity to increase SOC on maize and wheat lands to
improve crop yields.

**2.2 The interaction between SOC and N fertilizer on yield**
One of the key goals of sustainable intensification is to reduce the environmental impacts
of agriculture (Foley et al., 2011; Mueller et al., 2012). Nitrogen fertilization, while a
boon to yields, can cause environmental damages, such as eutrophication of waters and
increased soil emissions of nitrous oxide, a potent greenhouse gas (Vitousek et al., 2009).
Using our regression model, we asked whether there might be target N fertilizer addition
rates that suggest the possibility of maximizing yield per unit N applied by building SOC





and reducing N inputs. We wanted to see if yields converge at higher levels of SOC,
suggesting that crops are achieving sufficient nutrients through SOM and excess mineral
N is not necessary. Our analysis suggests that SOC is not directly substitutable for
mineral fertilizer (Fig. 2); however, at lower rates of N input ($\leq$50 kg N ha$^{-1}$), we found
that increasing SOC from 0.5 to 1.0% could potentially maintain current yields and
reduce fertilizer inputs by approximately half (50%). At higher rates of N input ($\geq$200 kg
N ha$^{-1}$), an increase from 0.5 to 2.0% SOC could potentially reduce N inputs by up to
70% per hectare (Fig. 4). These results suggest that building SOM in agricultural lands
may supply enough plant available nutrients to sustain crop yields while drastically
cutting back on N fertilizer inputs.

There was an interaction between SOC and N input, where at higher SOC

concentrations N input had a greater impact on yield (Fig. 2, Table 1). This may be
because higher SOC improves soil structure and water holding properties, resulting in
improved crop growth at a given level of N input (Powlson et al., 2011). Additionally,
soils receiving more N may have greater SOC because N increases crop yields, which can
increase the return of plant residues into the soil and potentially build SOC (Powlson et
al., 2011). However, if the relationship was simply an effect of greater inputs building
SOC, we should not have seen an interaction between SOC and N on yields (because
SOC should then just have been additively related to yield). Whatever the specific
explanation, the SOC by N interaction we detect suggests that a combination of both
building SOM and using targeted N applications could lead to potential increases in yield
(Fig. 4). Practices such as cover-cropping represent a strategy that can both increase N
supply and build SOM through biological N fixation and the return of high quality



residues (narrow C:N ratios) to the soil (Drinkwater et al., 1998). The combination of
both SOM improvement and targeted fertilizer input will likely be especially important
for degraded soils, which require a suite of organic and inorganic nutrients to help build
SOM and improve crop yields (Palm et al., 1997).
Gains in yield from fertilizer input leveled off at about 200 kg N ha$^{-1}$ y$^{-1}$ (Fig. 4),
meaning that optimum yields appear achievable, at least on average, with this fertilizer
input level and an SOC target concentration of 2%. Using this target N input rate, we
explored potential fertilizer reductions on agricultural lands using more than 200 kg N ha$^{-}$
$^{1}$ y$^{-1}$. We found that for lands receiving more than 200 kg N ha$^{-1}$ y$^{-1}$, current yields could
be maintained, while decreasing global N fertilizer inputs by 7% for maize and 5% for
wheat. It has been estimated that 25 to 30% of fertilizer N is exported to streams and
rivers, resulting in eutrophication (Raymond et al., 2012). Targeted reductions in the
application of fertilizer N on the order of magnitude our analysis suggests could then
prevent the annual export of as much as 3.73 million tonnes of N into inland waters,
which amounts to 10% of mineral fertilizer applied to maize and wheat lands.

**2.3 Exploring potential reductions in global yield gaps of maize and wheat**
With a majority of cultivated lands containing less than 2% SOC and a growing
imperative to build, restore, and protect SOC in agricultural soils (NSTC, 2016; FAO,
2008; NRCS, 2012), we used global gridded data sets coupled with our regression model
(Table 1) to examine the potential gains in yield and production if opportunities to
increase SOC are realized (Table 2). Although our model identified 2% as a global target
for SOC, we created regionally-specific SOC targets given the fact that achieving 2%





SOC in some soils (e.g. those of drylands) may be unachievable due to inherent
constraints of physical soil properties and climate (Stockmann et al., 2015). We created
region-specific SOC targets for each of the 18 agro-ecological zones (AEZ) defined by
the Food and Agriculture Organization (Ramankutty et al., 2007). We identified a target
of 2% for 14 of the 18 AEZs, and targets ranging from 1 to 1.5% for the remaining four,
arid AEZs (see Methods). These targets were in line with recent quantitative assessments
based on similar climatic classifications (Stockmann et al., 2015). We then used global
data sets on N input rate (Mueller et al., 2012), SOC (Hengl et al., 2014), pH (Hengl et
al., 2014), texture (Hengl et al., 2014) and aridity (Zomer et al., 2008) to fill each term in
our regression model to explore how increasing SOC concentrations to the regional
targets could potentially affect global yields. We found that increasing SOC
concentrations to the defined targets has the potential capacity to increase average yields
on a per hectare basis by 10±11% (mean±SD) for maize and 23±37% for wheat. These
gains in yield translate to a 5% and 10% increase in the global annual tonnes produced of
maize and wheat, respectively (Table 2). These increases in production would close 32%
of the global yield gap for maize and 60% of the gap for wheat (Fig. 5a, b).

These yield gap results represent an exploration of potential "best case" impacts

of increasing SOC concentrations. We recognize there are inherent and logistical
challenges to building SOM in agricultural soils. For instance, soil characteristics such as
texture have a large effect on SOC content because sandier (rather than more clay rich)
soils have less surface area to stabilize SOC (Cotrufo et al., 2013), and so hold much less
water and nutrients than clay-rich soils (Johnston et al., 2009). Maintaining SOC contents
in sandy soils may require more frequent additions of organic amendments because these



soils do not have the surface area to retain nutrients, moisture, and to stabilize SOC
(Lehmann and Kleber, 2015). Furthermore, different regions and climate types face
different imperatives for building SOM. In the mid-western United States, for instance,
building SOM may be a good strategy to reduce fertilizer inputs and irrigation needs;
whereas in sub-Saharan Africa, building SOM may be critical for drought protection and
nutrient provision. Notably, high SOM values are not common in dryland environments
(for our dataset, mean SOC = 0.9% for dryland climates versus 1.4% SOC for mesic
soils), and building and maintaining SOM in arid zones is typically hindered by the lack
of organic matter to return to soils (Rasmussen et al., 1980). On a positive note, however,
our analysis suggests that increases in SOC in drylands, for example, from 0.5 to 0.8%,
could potentially increase yields by 10%, likely due to impacts on water retention.

Whereas we did use lower SOC targets (ranging from 1.0 to 1.5%) for the arid

AEZs in our analysis, the majority of data used for our analysis is from the more
temperate and tropical humid zones (Fig. S2). To bolster and/or refine SOC targets, our
correlative analysis needs to be supplemented with well-replicated experimental studies
incorporating different management strategies across multiple soil and climate types to
develop SOC-yield relationships that can be applied to the specific set of local farm
conditions. Further, these studies should ideally report data related to soil texture and
mineralogy, nutrient management, and paired SOC-yield observations with SOC taken to
meaningful depths, such as those that represent plant-rooting depth. These experimental
studies will help generate information that practitioners can use to inform management by
taking into account the inherent and logistical challenges to building SOM in agricultural
soils.




**3 Conclusions**

Despite uncertainties and calls for further research into how SOM affects agricultural
performance (Cassman, 1999; Herrick, 2000; Oldfield et al., 2015), policy for sustainable
intensification already widely supports the merits of increasing SOM in agricultural lands
(FAO, 2008; NRCS, 2012). The purported benefits include improved yields, increased
resilience, and decreased inputs of fertilizer and irrigation water. However, although
consensus exists around the importance of SOM to soil health, translating SOM policy to
practice is hindered by the lack of a predictive capacity for SOM target setting to inform
management efforts focused on yield and reducing fertilizer and irrigation (Chabbi et al.,
2017; Herrick, 2000; NRC, 2010). Our analysis helps establish a quantitative framework
for SOC targets that achieve measurable agricultural outcomes as part of sustainable
intensification efforts. It quantifies the potential effect size of SOC on yield while also
accounting for climate, soil, and management variables that influence crop yield. We find
that greater concentrations of SOC are associated with greater yields up to an SOC
concentration of 2%. With two thirds of global maize and wheat lands having SOC
concentrations of less than 2%, there seems significant opportunity to increase SOC to
reduce N inputs and potentially help close global yield gaps.

**4 Methods**

Our approach consisted of a two-stage process. In the first stage, we assembled published
empirical data from studies that reported both SOC and yield data for maize and wheat.
From this meta-dataset, we then quantified how both SOC concentrations and N input



rates are related to yields, in the context of spatial variation in climatic, management, and
soil co-variables. In the second stage, we used globally-gridded data sets to extract values
for the factors we investigated in the first stage for global lands where maize and wheat is
produced. Using the regression relationship developed from the published empirical data
compiled under the first stage, we then estimated how an increase in SOC concentrations
up to target thresholds we identified (ranging from 1 to 2% depending on agro-ecological
zoning) affected global yield potentials. Finally, we used an N input threshold identified
through our regression analysis (200 kg N ha$^{-1}$ yr$^{-1}$) to calculate potential N reductions on
global maize and wheat lands.

**4.1 Data collection**

In the first stage of our approach, we searched the database Web of Science (Thomson
Reuters) in January 2016 and again in October 2016 using the following topic search
terms: "soil organic matter" OR "soil organic carbon" OR "soil carbon" OR "soil c"
AND "yield" OR "crop yield" OR "productivity" AND "agricult*." We restricted the
initial search to articles published in English between 1980 through December 2015, and
excluded conference proceedings; the second search captured articles published in 2016.
The initial search resulted in 1,384 articles and the second 169 articles (Fig. S1). For each
citation, we reviewed titles and abstracts to select articles that met the following criteria:
experimental field studies whose abstract included information on yield and SOC for
systems growing wheat and/or maize. This initial screening resulted in 523 records for
which we assessed the full text. We assessed these records for eligibility based on
inclusion of data on crop yield, SOC, and N fertilizer rates for each observation. For



inclusion within our analysis, it was essential that studies reported paired SOC and yield
data. Furthermore, we required SOC concentrations (as opposed to stocks). Studies did
not meet our criteria for inclusion if they reported SOC stocks with no corresponding data
on bulk density to convert into concentrations; and also if they reported baseline SOC
concentrations as opposed to experimental SOC concentrations that we could pair with
yield data. In addition to our literature search, we also contacted authors to see if they
were willing to include raw data within our database. This resulted in three datasets
(Adiku et al., 2009; Birkhofer et al., 2008; Kautz et al., 2010). Finally, we consulted the
recently published database by the Swedish Board of Agriculture that is a key repository
of peer reviewed literature focusing specifically on studies (735 in total) related to the
effects of agricultural management on soil organic carbon (Söderström et al., 2014). We
explored this database to find studies from regions that were under-represented within our
literature search (e.g. the southern hemisphere). This resulted in a search of 55 studies to
see if they met our criteria for inclusion. We scanned each paper to see if they included
SOC data paired with matching yield data. From these papers, we extracted data from 12
studies, which resulted in an additional 52 data points. We encountered limitations
similar to our initial search: Namely, SOC and yield data were not paired, studies
included only baseline SOC concentrations, or SOC stocks were reported without any
corresponding bulk density data to convert into concentrations. Overall, our data set
included 840 individual observations from 90 articles covering sites across the globe
(Fig. 1) (Adiku et al., 2009; Agegnehu et al., 2016; Albizua et al., 2015; Alijani et al.,
2012; Araya et al., 2012; Atreya et al., 2006; Bai et al., 2009; Bedada et al., 2014;
Bhardwaj et al., 2011; Bhattacharyya et al., 2015; Birkhofer et al., 2008; Boddey et al.,



2010; Boulal et al., 2012; Bremer et al., 1994; Calegari et al., 2008; Campbell et al.,
2007; Castellanos-Navarrete et al., 2012; Celik et al., 2010; Chen et al., 2015; Chirinda et
al., 2010; Cid et al., 2014; Costa et al., 2010; D'Hose et al., 2014; Datta et al., 2010;
DeMaria et al., 1999; Diacono et al., 2012; Grandy et al., 2006; Guo et al., 2012; 2009;
He et al., 2011; Hossain et al., 2016; Hu et al., 2015; 2014; Kaihura et al., 1999;
Karbozova Saljnikov et al., 2004; Kautz et al., 2010; Kazemeini et al., 2014; Kucharik et
al., 2001; Larsen et al., 2014; Lebbink et al., 1994; Leogrande et al., 2016; Li et al., 2015;
Liu et al., 2014a; 2016; 2014b; 2014c; López-Garrido et al., 2014; Lu et al., 2016; Ma et
al., 2012; 2016; Madejón et al., 2001; Mandal et al., 2013; Masto et al., 2007; Mikanová
et al., 2012; Mishra et al., 2015; Mupangwa et al., 2013; N'Dayegamiye, 2006; Niu et al.,
2011; Njoku and Mbah, 2012; Paul et al., 2013; Qin et al., 2015; Quiroga et al., 2009;
Sadeghi and Bahrani, 2009; Saikia et al., 2015; Scalise et al., 2015; Seremesic et al.,
2011; Singh and Dwivedi, 2006; Singh et al., 2016; Sisti et al., 2004; Soldevilla-Martinez
et al., 2013; Spargo et al., 2011; Šimon et al., 2015; Tejada et al., 2016; Tiecher et al.,
2012; van Groenigen et al., 2011; Vieira et al., 2007; 2009; Wang et al., 2015; 2014a;
2014b; Wortman et al., 2012; Wu et al., 2015; Yang et al., 2013; 2015a; 2015b; Yeboah
et al., 2016; Zhang et al., 2015; 2009; 2016; Zhao et al., 2016). Where necessary, we
extracted data from manuscript figures using GraphClick Software (v. 3.0.3,
http://www.arizona-software.ch/graphclick/).

Studies that presented individual data points recorded over multiple years were

included as well as studies that averaged both yield and SOC data over multiple years. To
avoid over-representation of studies that included data points recorded for both yield and





SOC over multiple years (>10 y), we took observations from the beginning, middle, and
last year of the study.

**4.2 Data compilation**
For each extracted observation, we compiled the following information: latitude,
longitude, year of data collection, crop type, yield, SOC or SOM, depth of SOC or SOM
measurement, N fertilization rate, P fertilization rate, soil pH, texture, and whether or not
crops were irrigated. We used SOC (as opposed to SOM) for our analysis given that SOC
is a common proxy for SOM. Carbon, as an element that is easily identified and
measured within soil, is thought to comprise ~50-60% of SOM and is commonly reported
in the literature (Pribyl, 2010). When SOM was reported, we converted it to SOC by
dividing the value by 1.724 (Cambardella et al., 2001). Different studies reported SOC
concentrations to different depths, which ranged from 0-5 cm to 0-30 cm, with the
majority of studies reporting SOC to 0-20 cm. When studies reported SOC to multiple
depths, we averaged SOC values across depths to 30 cm. If no information on irrigation
was provided, we scored the observation as rain-fed. Soil texture and pH were not
reported for every study; 79% of included studies reported pH, and so we used the
study's latitude and longitude to extract these data using ISRIC SoilGrids (Hengl et al.,
2014) to fill in the missing pH values. Texture was reported for about half (49%) of
included studies, and so we used coordinates to pull these data from SoilGrids as well
(Hengl et al., 2014). We also used latitude and longitude to obtain an "aridity index"
through the CGIAR-CSI database (Zomer et al., 2008).



### 4.3 Data analysis


We used a linear mixed model (LMM) to analyze the observations we extracted from the
literature. Our model included SOC, N fertilizer rate, crop type (maize or wheat, coded as
a binary variable), irrigation (coded as a binary variable), aridity index, latitude, pH, and
texture (% clay) as fixed effects. To account for any spatial and temporal correlation
among the studies, we nested year within study as random effects (Bolker et al., 2009).
The LMMs were fit with a Gaussian error distribution in the "lme4" package for the "R"
statistical program (version 3.3.1), using the "lmer" function. The first stage of our data
analysis was to test the data distributions. We removed data points with N fertilization
rates >600 kg N ha$^{-1}$ (4 data points) and yields >18 t ha$^{-1}$ (2 data points) since these
represented outliers for our dataset (being beyond 3 times the inter-quartile range of the
meta-dataset) and are not representative of on-farm management practices or outcomes.
Our final model was based on 834 observations across 90 studies. We added quadratic
terms for both SOC and N input rate since these variables exhibited a nonlinear
relationship with yield. The square-root of the variance inflation factors (vif) was <2 for
all factors when included as main effects, indicating that collinearity was low among all
variables. As would be expected, there was a correlation between SOC and its quadratic
term and N input rate and its quadratic term. We calculated the $r^2$ values for our model
following Nakagawa and Schielzeth (2013) to retain the random effects structure. The $r^2$
of our model was 83% for the full model, with the fixed effects explaining 42% of
observed variance within our data set.

We based the choice of factors for inclusion in our model on the approach of

Hobbs et al. (2006), by only investigating factors where biological mechanism as to their



influence on yield is firmly established and where we were interested in their effect sizes
relative to one another. Also following Hobbs et al. (2006), we did not carry out model
selection. Operationally, there is substantial subjectivity and lack of agreement in model
selection approaches, with different decisions leading to markedly different conclusions
as to the influence of different factors. Instead, coefficients are generally most robust
when all terms are retained in a model, assuming that inclusion of each is biologically
justified.

To examine the effects sizes of the factors on yield, we took two approaches.

First, we compared the size of the standardized coefficients, where standardizing
involved subtracting the mean of the factor from each observed value and dividing by
two standard deviations (Gelman, 2008). Dividing by two standard deviations is useful
when binary predictors are included within regression models (in our case, crop type and
irrigation are coded as a binary predictors). This way, continuous and binary variables all
have a mean of 0 and a standard deviation of 0.5 (Gelman, 2008). This accounts for the
fact that the factors were measured on different unit scales (Table 1). Second, we
examined the influence of changing SOC concentration or N fertilization rates on yield.
To do this, we used the regression relationship derived from our statistical model, held all
other factors at a constant value (e.g. the mean of all observations for that factor), and
systematically varied SOC or N fertilization across the range of values we extracted from
the literature. For SOC, this meant varying SOC values from 0.1 to 3.5% to estimate
changing yield of rain-fed maize or wheat as SOC concentrations were increased (Fig. 2).
For N fertilization, we varied N input rates from 0 to 300 kg N ha$^{-1}$ for rain-fed maize or
wheat at different SOC concentrations (Fig. 4). When these factor-yield relationships

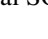
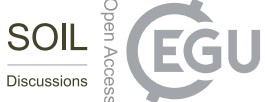

were plotted, we identified threshold values where yield became minimally responsive to
SOC or N fertilization as the point where the slope of the relationship became <0.25 (for
SOC) and < 0.002 (for N fertilization).

**4.4 Global extrapolations**

We used the regression relationship developed in the first stage of our approach to predict
how building SOC concentrations would potentially affect global crop yield averages. To
obtain values for each of the factors in our regression model at a global scale, we used
globally gridded data products. Global SOC, pH, and texture data were taken from ISRIC
SoilGrids (Hengl et al., 2014) at a 10-km grid cell resolution to match the spatial grain for
maize and wheat yields and N fertilization data, which we obtained from the EarthStat
product (Monfreda et al., 2008; Mueller et al., 2012). SoilGrids has multiple layers for
SOC concentrations, and we used the 0-15 cm layer as the average depth to which SOC
was reported for our dataset was 0-20 cm. The aridity index was obtained from the
CGIAR-CSI database (Zomer et al., 2008). We used the resulting global dataset to
explore the potential impact of increasing SOC (up to regionally identified threshold
levels ranging from 1 to 2%) on yield for lands across the globe where maize and wheat
are produced.
To establish regionally appropriate SOC targets, we classified maize and wheat
producing areas by their agro-ecological zones (AEZ). The Food and Agricultural
Organization have 18 zones defined on the basis of combinations of soil, landform, and
climatic characteristics (Ramankutty et al., 2007). For each AEZ, we examined the
distribution of SOC in areas classified as naturally vegetated (e.g. not in urban or

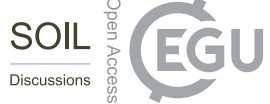

agricultural land uses). We did this by stacking two GIS raster layers of SOC (SoilGrids)
and land use (Friedl et al., 2010), excluding agricultural and urban land use
classifications. We then extracted SOC data for each AEZ using a shape file outlining the
geographical extent of each AEZ (Ramankutty et al., 2007). Examining the distribution
of SOC across each AEZ, we identified targets based on the mean SOC value within each
zone. All but four zones had means greater than 2% SOC, so we set target values for
those zones at 2%. Mean SOC concentrations were lower for the more arid zones and so
we set those targets to 1% for AEZ 1 and 1.5% for AEZ zones 2, 3, and 7. Recent
analysis of global SOC concentrations across globally defined Ecoregions shows mean
values of SOC at or greater than 2% for all regions except land classified as desert and
xeric shrubland (Stockmann et al., 2015).

Prior to our global extrapolations, we performed a suite of data checks. We

wanted to ensure that global yields predicted using our regression model were
comparable to those from EarthStat. These checks helped validate the strength of our
extrapolations. Firstly, we explored the range of variation in variables from experimental
data used to generate our model as well as the range of global variation in variables we
project across. The range of our regressors encompasses the range of global variation,
except for aridity, in which case 4.6% percent of our projections fall in grids that have
axis conditions outside of our range of measurements. These values fall in extremely arid
systems, with aridity values of less than 0.1. In these extremely arid zones, we do make a
point to use much lower target SOC values, recognizing that achieving 2% SOC in these
very arid areas is not very likely. Secondly, using our regression model to predict global
yields for both maize and wheat (separately), we first removed all values from the





analysis that had predicted yields of less than 0 because negative yields are not possible.
This amounted to 0.004% of the total predictions for maize and 0.15% for wheat. For
clarification, we refer to predictions from our regression model as "predicted" or "model-
predicted." We then calculated the proportional difference between model-predicted and
globally gridded yield data from EarthStat. We dropped all cells for which the
proportional difference between predicted and gridded data was >3-times. This threshold
represents the mean ± half of the standard deviation for the distribution of the
proportional difference between predicted and EarthStat yield data. This amounted to
14% of cells for maize and 7% for wheat. The mean proportional difference between
predicted and gridded data was 0.85±0.91 for maize (Fig. S3b) and 0.45±0.87 for wheat
(Fig. S4b). The correlation between predicted and gridded data was r=0.73 for maize
(Fig. S3c) and r=0.38 for wheat (Fig. S4c). We also visualized overlap in the distribution
of model-predicted and gridded data. Model-predicted maize yield had a global mean of
4.66±1.84 t ha$^{-1}$ and EarthStat had a global mean of 3.34±2.62 t ha$^{-1}$ (Fig. S3a). Model-
predicted wheat yield had a global mean of 3.18±1.66 t ha$^{-1}$ and EarthStat had a global
mean of 2.43±1.58 t ha$^{-1}$ (Fig. S4a).

We also compared the distribution of EarthStat yield data with observed yield

data from the studies included in our analysis. We found that correlation (r values)
between the gridded and collected data was 0.56 for maize and 0.39 for wheat. Average
observed maize yield was 5.61±3.32 t ha$^{-1}$ and wheat yield was 4.02±2.11 t ha$^{-1}$
(mean±SD). EarthStat maize yield, again, was 3.34±2.62 t ha$^{-1}$ and wheat yield was
2.43±1.58 t ha$^{-1}$. These differences between predicted and EarthStat yield averages are
likely due to the fact that EarthStat data is based on regional census data, incorporating





much more variability in terms of management practices and skill than experimental field
studies.

After the data checks, we then used our model to extrapolate global yield

potentials of maize and wheat given increases in SOC. We masked EarthStat production
and cultivated area data layers for maize and wheat for cells that had SoilGrids SOC
concentrations of >2%. We compared the subsetted data (i.e. cultivated lands with < 2%
SOC) with the original data layers to determine the fraction of global maize and wheat
production and cropland that is on soils with less than 2% SOC. We used this subsetted
data along with our regression model to predict yields at current SOC levels. As stated
above, we used EarthStat, ISRIC SoilGrids, and CGIAR-CSI data layers to fill in the
values for each of the factors in our regression model. This new data layer was used as a
baseline with which to compare to potential gains in yield with an increase to SOC target
values. This created a second data layer with model-predicted yields given an increase in
SOC. We calculated the percentage increase in yield between these two layers (the
baseline and the improved-SOC layer) and multiplied this by EarthStat yield and
production data to determine potential gains in maize and wheat yields and production
(Table 2). We then used EarthStat yield gap data to see how such an increase in SOC
would reduce projected yields gaps. Using the new yield data layer (with yields at SOC
target values), we calculated the proportion of EarthStat yield gaps that was reduced for
both maize and wheat.

Finally, we used data on global N use (EarthStat) to explore potential reductions

in fertilizer use for both maize and wheat, separately. We used a value of 200 kg N ha$^{-1}$ y$^{-}$
$^{1}$ as our N input threshold, as this is the value from our regression model at which gains



in yields level off. We created a new data layer for those areas that have N input rates
greater than 200 kg N ha$^{-1}$ y$^{-1}$. We then calculated the potential N reductions, in tonnes,
by multiplying this new data layer by EarthStat cultivated maize and wheat lands,
separately. Finally, we divided the potential reduction in N input (in tonnes) by total N
input (in tonnes) as provided through the EarthStat data product.

**Data availability**
The dataset generated and analyzed during the current study is available through the
KNB repository: https://doi.org/10.5063/F1RV0KWK

**Author contributions**
EEO, MAB, and SAW conceived of the study. EEO and SAW performed data analysis.
EEO wrote the first draft of the manuscript. All authors contributed to data interpretation
and paper writing.

**Competing Interests**
The authors declare that they have no conflict of interest.

**Acknowledgements**
Thanks to Samuel Adiku, Klaus Birhofer, and Tim Kautz for their contributions of data.
Thanks also to the SNAPP working group on 'Managing Soil Carbon' for their support, as
well as Deborah Bossio, Indy Burke, Jon Fisher, Cheryl Palm, Pete Raymond, and the
Bradford Lab Group for comments on earlier drafts.



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





**Table 1. Modeled regression coefficients with standard errors, standardized**
**coefficients, and *P* values for our regression model.**

| Variable | Unstandardized Coefficients | Standardized Coefficients | *P* value |
|---|---|---|---|
| Intercept | -1.62 ± 1.71 | 5.59 ± 0.18 | 0.34 |
| SOC | 1.78 ± 0.59 | 1.44 ± 0.30 | 0.003 |
| $SOC^2$ | -0.46 ± 0.18 | -0.66 ± 0.27 | 0.013 |
| N input | 0.018 ± 0.0014 | 2.71 ± 0.15 | < 0.00001 |
| N input$^2$ | -0.000039 ± 0.0000036 | -1.64 ± 0.15 | < 0.00001 |
| Irrigation | 0.77 ± 0.35 | 0.77 ± 0.34 | 0.027 |
| pH | 0.053 ± 0.18 | 0.12 ± 0.42 | 0.77 |
| Aridity | 0.16 ± 0.51 | 0.12 ± 0.41 | 0.76 |
| Crop Type | 1.55 ± 0.15 | 1.54 ± 0.15 | < 0.00001 |
| Clay (%) | 0.013 ± 0.014 | 0.29 ± 0.31 | 0.36 |
| Latitude | 0.055 ± 0.016 | 1.40 ± 0.41 | 0.00077 |
| SOC*N input | 0.0039 ± 0.00099 | 0.96 ± 0.25 | 0.00010 |

The output of our linear mixed effect model (n=834). The full model explained 83% of
observed variability within the data set with fixed effects (included in the table)
accounting for 42% of the variability. Standardized coefficients allow for direct
comparison of the relative effect size of each modeled variable despite different scales on
which the variables are measured. For example, crop type's effect on yield is two-times
greater than that of irrigation. Crop type was coded as a binary variable with 0 for wheat
and 1 for maize. Irrigation was also coded as a binary variable with 0 for no irrigation and
1 for irrigation.





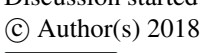



**Table 2. Scenarios for increases in yield and reductions in N input with an increase in SOC concentration to target values.**

| Scenario | Crop | Global Yield Average (t ha$^{-1}$) | Increase in production (Mt) | Nitrogen input (Mt N ha$^{-1}$) |
|---|---|---|---|---|
| Current condition | Maize | 3.34 ± 2.62 | NA | 17.24 |
| | Wheat | 2.43 ± 1.58 | NA | 33.07 |
| Increase SOC to target concentrations | Maize | 3.93 ± 3.08 | 29.96 | 15.96 |
| | Wheat | 3.17 ± 2.06 | 55.41 | 32.04 |

Values (mean±SD) represent current EarthStat yields and projected gains in yield and production resulting from an increase in SOC concentration to target values for each agro-ecological zone (targets ranged from 1.0 to 2.0%). We used our regression model to determine potential gains in EarthStat yield and reductions in EarthStat N input. Global yield averages represent tonnes produced per unit land area, whereas production represents tonnes of maize and wheat produced globally.





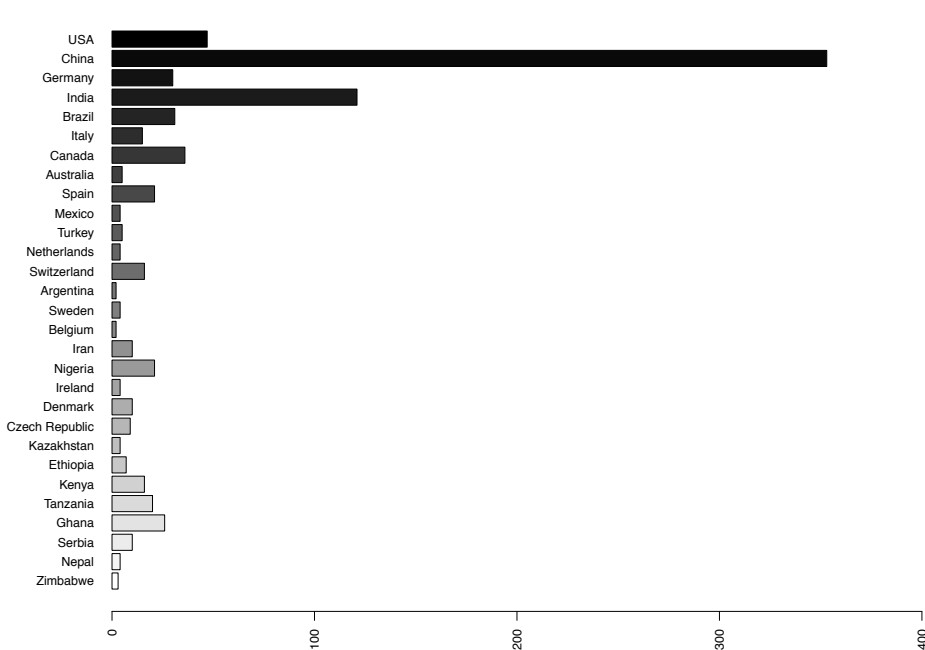

1108

**Figure 1: Distribution of data points by country.** Countries are ordered by Gross

Domestic Product (GDP) in order from largest (top) to smallest (bottom). The data set

used for this study contains a total of 840 individual observations from 29 different

countries.

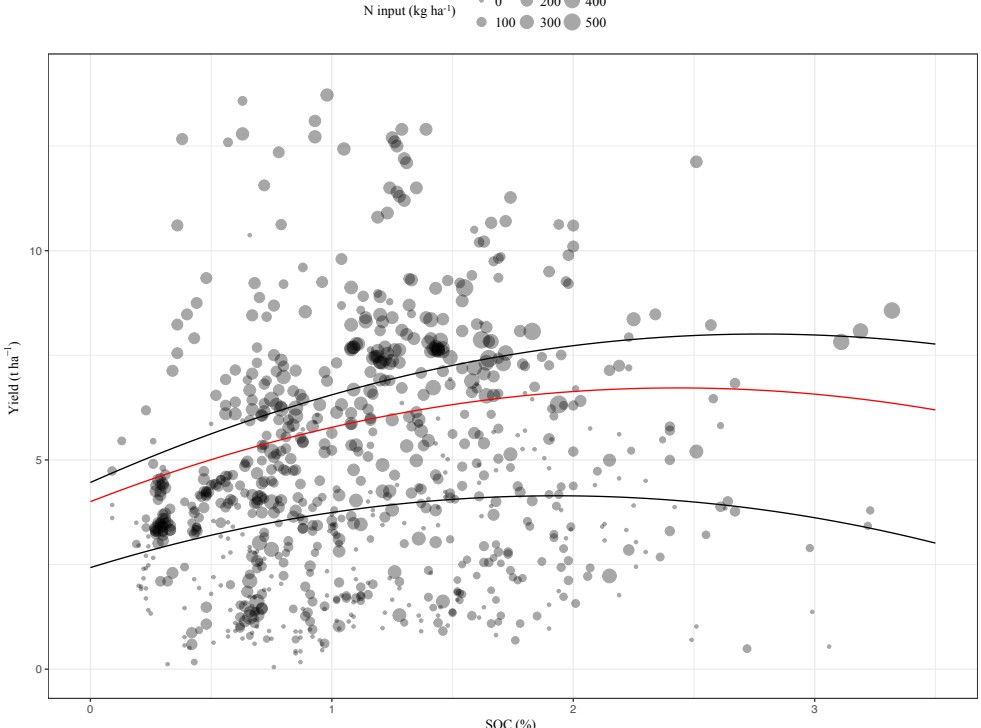

1113

**Figure 2: Relationship between SOC and yield of maize and wheat for published**

**studies.** The regression lines are modeled yields (i.e. effect sizes) for rain-fed maize

using observed means of our meta-dataset for aridity, pH, texture, and latitude at different

N input rates. We varied SOC (x-axis) across the range of values extracted from the

literature. The red line represents the mean N input rate (118 kg N ha$^{-1}$ y$^{-1}$) across all

studies, with the bottom line representing 0 inputs of N and the top line representing 200

kg N ha$^{-1}$ y$^{-1}$. For the raw data points, N input is mapped as a continuous variable across

its range from 0 (smallest circles) to 500 kg N ha$^{-1}$ y$^{-1}$ (largest circles). Note that the

observed scatter of the individual observations is an outcome of the fact yield is

controlled by multiple factors (Table 1), and therefore the regression lines isolate just the

potential effect of SOC with all other factors held constant.



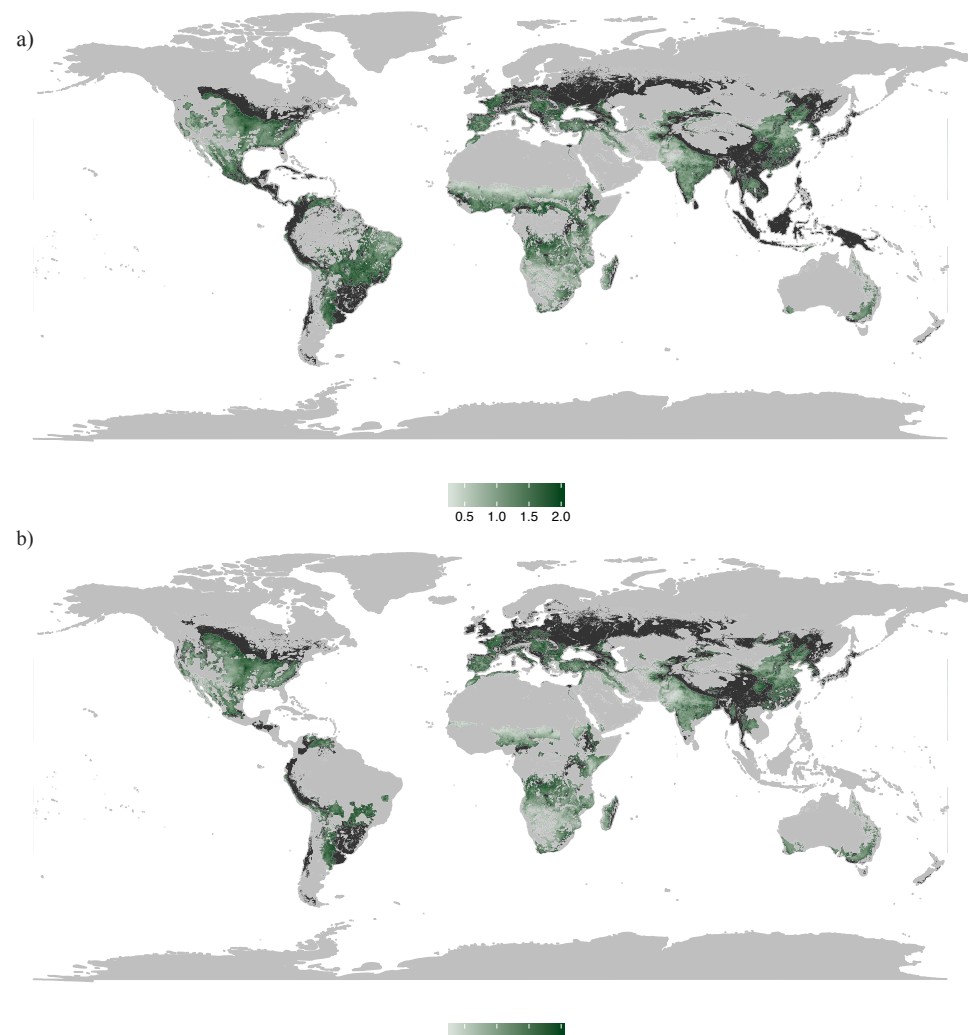

1125

**Figure 3: Global maize and wheat lands with less than 2% SOC.** Cultivated (a) maize

lands and (b) wheat lands on soils with SOC contents less than 2%. Approximately two

thirds of all maize (61%) and of all wheat (64%) producing areas are on soils with less

than 2% SOC. Black areas on the maps are cultivated maize and wheat lands that have

concentrations over 2% SOC. Yield data is taken from EarthStat and SOC data is taken

from ISRIC-SoilGrids.



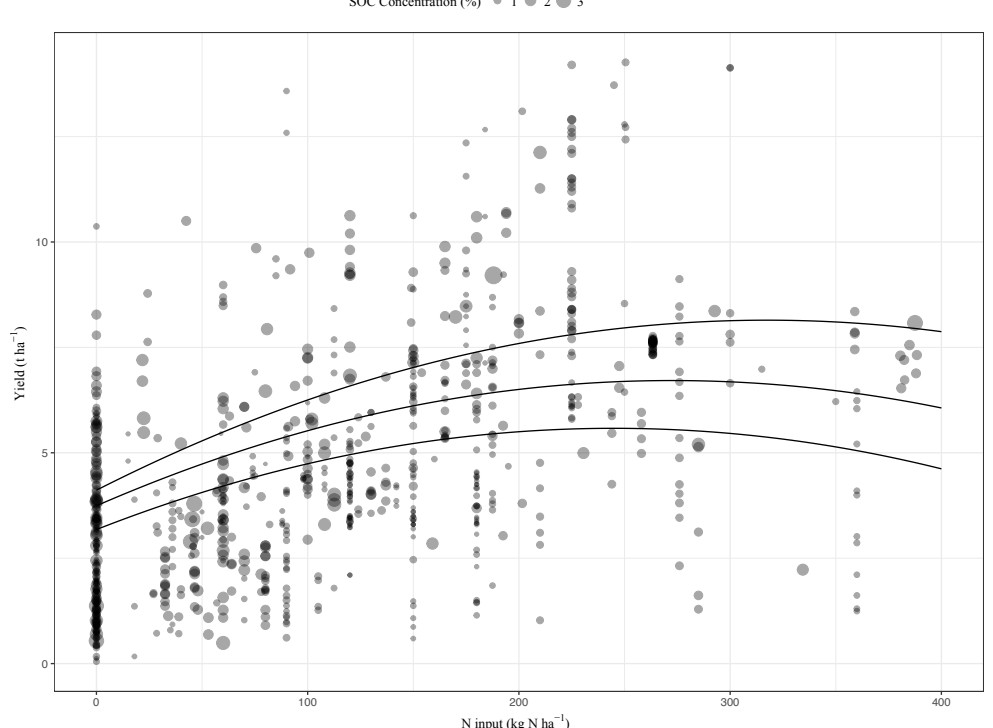

1132

**Figure 4: Potential reductions in nitrogen fertilizer with an increase in SOC**

**concentration.** The lines on the graph represent varying SOC concentrations, 2.0%,

1.0%, and 0.5% SOC from top to bottom. These lines are plotted on top of the

observations from our data set with SOC mapped as a continuous variable across its

range from 0.1% (smallest circles) to 3.0% (largest circles). Our model shows that

keeping yield constant by increasing SOC contents allows for potentially significant

reductions in N input (e.g. the same yield is achievable with 0 N input and 2% SOC, as

with 65 kg N ha$^{-1}$ y$^{-1}$ and 0.5% SOC). Recognizing that the 0 N input values may

influence the modeled relationship, we analyzed data excluding these values. The

qualitative patterns remain the same if the 0 N input values are excluded from the

analysis; and while the absolute quantitative patterns shift slightly, the general trends



remain intact.

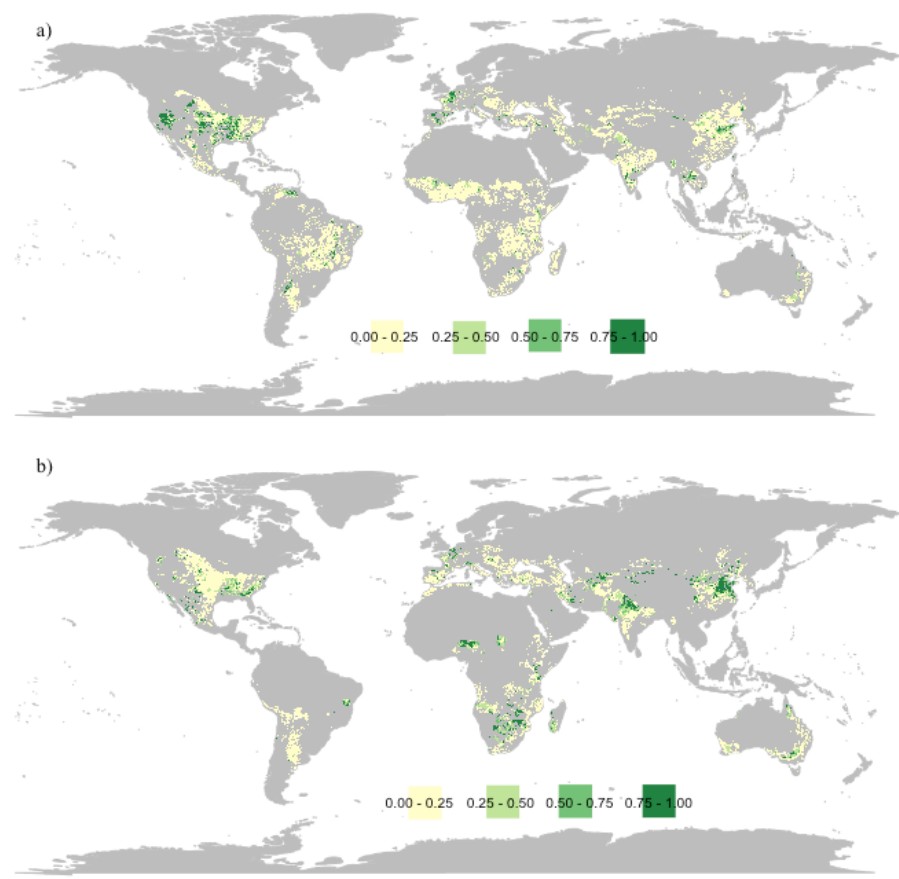


**Figure 5: Percentage closure of yield gap for (a) maize and (b) wheat given an**
**increase in SOC concentration to target values for each AEZ (ranging from 1-2%).**
Modeled gains come from our regression relationship between SOC and yield and
applying it to EarthStat yield gap data. Doing so determines the potential increase in yield
and therefore projected reductions in yield gaps for maize and wheat.