# Peer review of "Global meta-analysis of the relationship between soil organic matter and crop yields"

_SOIL, 2018_

## Referee Comment (RC1) · Anonymous Referee #1 · 14 Sep 2018

The authors use a global data set on maize and wheat yields together with soil and other environmental variables to derive statistical relationships between SOC and yield. The overall value of the study is appreciated. The interpretation of the data and observed relationships is, however, going too far because direct evidence for the postulated effects, as it could be derived from long-term experiments at different SOC levels, cannot be derived and many other influencing factors were ignored.

Title and abstract. In both, SOM is described as the key variable but the study relies on SOC data. This should be reflected in the title and the abstract. This already touches a more fundamental problem – the study does not provide mechanistic insight as to why higher SOC results in higher yields. More SOC is often obtained using more organic inputs, i.e., more macro- and micro-nutrients bound to SOM. A second issue here,

related to the first one is that, correctly, a higher SOC concentration might reduce the amount of N needed as fertilizer to get the same yield, but it is not discussed how much more N must be fertilized to reach the higher SOC level.

L. 96 and methods. It is not clear why authors only used aridity and latitude as variables related to climate. Yields are strongly related to rainfall and temperature, which are easily available variables.

L. 121. More recent literature suggests that higher yield is not coming along with higher plant residue inputs (e.g., Hirte et al. 2018 Agriculture Ecosystems Environment 265).

L. 141. Authors argue that two thirds of maize and wheat cultivation takes place on soils with less than 2 % SOC. What is, for comparison, the average % SOC of croplands worldwide? Are these two staple crops planted on particularly C-poor soils?

L. 160. Are the authors aware of any long-term field experiment where an increase from 0.5 to 2 % SOC has been observed? This seems unlikely to me. Even a doubling (previous sentence) is ambitious. The following argumentation, that higher SOM soils may supply enough plant available nutrients to sustain crop yields with drastically cutting back N fertilizer input overlooks that these are typical situations of SOM decline, as observed in many long-term experiments, where plant productivity can be maintained at low nutrient input rates only because of SOM decline and the associated release of organically bound nutrients.

L. 193 ff. The first para in section 2.3. belongs largely to the method section and is partially a repetition of that.

L. 215. It is not clear where the yield gap comes from – how was it calculated, was it taken from the literature? Clarification needed.

L. 302. Authors refer to Söderström et al. 2014. I looked up that reference where I could not find a database as key repository but rather a research approach. Should be clarified.

L. 352. I suggest to use three classes: rainfed, irrigated, unknown.

L. 353. Filling data gaps for soil pH and texture for experimental sites by a global database may introduce large errors and, potentially, biased estimates, given that these soil properties vary much over short distances. I suggest to either exclude those variables as explanatory ones or to ask authors of the studies to provide those data for their sites. Alternatively, these parameters can be categorized and used as categorical variables.

Table 2. I suggest to add a percentage increase in production from an increase in SOC to the table to make the global yield average and the increase in production comparable to each other.

Figure 2. Not clear why the figure relates to maize and yield in line 1114 whereas the caption in line 1115 refers to maize only.

Figure 5. The figure is interesting but results would better be presented as percentage increase in yield, and not as percentage closure of yield gap. The yield gap itself is prone to large uncertainty, both in extent and possible reasons, and these uncertainties are not explicitly included.

Figure 4. The provided interpretation of this results ignores the fact that building up additional SOC requires additional N.

---

## Author Comment (AC1) · 19 Sep 2018

The authors use a global data set on maize and wheat yields together with soil and other environmental variables to derive statistical relationships between SOC and yield. The overall value of the study is appreciated. The interpretation of the data and observed relationships is, however, going too far because direct evidence for the postulated effects, as it could be derived from long-term experiments at different SOC levels, cannot be derived and many other influencing factors were ignored.

Title and abstract. In both, SOM is described as the key variable but the study relies on SOC data. This should be reflected in the title and the abstract. This already touches a more fundamental problem – the study does not provide mechanistic insight as to why

higher SOC results in higher yields. More SOC is often obtained using more organic inputs, i.e., more macro- and micro-nutrients bound to SOM. A second issue here, related to the first one is that, correctly, a higher SOC concentration might reduce the amount of N needed as fertilizer to get the same yield, but it is not discussed how much more N must be fertilized to reach the higher SOC level.

Response: We appreciate the reviewer's comments and believe we can fully address them in a revision. The first major concern is that we do not provide mechanistic insight as to why SOM (or SOC) would increase yield. We believe that the mechanisms between SOM/SOC and crop yield have been well established, but poorly quantified. For instance, we would expect SOC to be associated with greater cation exchange capacity for the exchange of micronutrients and greater water holding capacity. SOC, because it is the majority constituent of SOM, is also highly correlated with macro-elements contained in SOM. The contribution of our project is not to tease apart the relative importance of the separate mechanisms by which SOM/SOC operates, though we do believe this would be a very important, but challenging, project. Instead, our aim is to establish relationships at broad scales between SOC and yield to provide better quantification of this relationship for policy initiatives such as the recently launched Global Soil Health Challenge. This has been identified as a critical knowledge gap among producers, policy makers, and researchers alike (Adhikari and Hartemink, 2016; Chabbi et al., 2017; Hatfield et al., 2017). For instance, the U.S. National Research Council stated in their 2010 report on sustainable agriculture that "measures of [SOM] are a cornerstone of most sustainability and soil quality assessments...However, the numerical level that would be considered good, or what change in [SOM] levels constitutes a significant functional change, has not been established (NRC, 2010)." Our paper is an attempt to answer that call, and we will make this clear in a revision.

Secondly, the reviewer raises concerns related to the challenge of using observational differences across broad spatial scales to test SOC-yield relationships. We note here and will make clear in our revision that the differences in soil carbon observed in our

data set are from experimental plots capturing long-term differences in SOC within a given site. Specifically, our data capture differences within SOC in experimental plots largely driven by management interventions related to inputs (e.g. compost, fertilizer, manure, crop residues) and tillage (e.g. no-till versus till). We capture these site-specific differences in management with site-level random intercept terms.

Regarding our analysis of potential fertilizer reductions: we recognize that a combination of both organic and inorganic nutrients will be necessary to help build SOM and improve crop yields (lines 179-180). We will highlight this further in a revised manuscript and stress that building SOM and cutting back on N fertilizer will require achieving an agricultural N balance where SOM-N mineralization accounts for the reductions in mineral fertilizer. This will depend on the amount and C:N ratios of inputs used in specific agricultural systems. We also note and will provide relevant citations that this may be prove especially challenging in smallholder systems where there is often lack of access to and insufficient quality of organic inputs (Giller et al., 2009; Palm et al., 2001).

L. 96 and methods. It is not clear why authors only used aridity and latitude as variables related to climate. Yields are strongly related to rainfall and temperature, which are easily available variables.

We chose to use aridity since it is a variable that is expressed as a function of precipitation, temperature, and potential evapo-transpiration. We will include this information and relevant citation in a revised manuscript (Trabucco 2009). We did initially include rainfall and temperature variables in our statistical model, but since they were highly correlated, we chose to leave them out and include aridity since it is derived from temperature and precipitation data. The use of aridity has been used in other large-scale yield studies (Pittelkow et al., 2014).

L. 121. More recent literature suggests that higher yield is not coming along with higher plant residue inputs (e.g., Hirte et al. 2018 Agriculture Ecosystems Environment 265).

This is a good point, and one we highlight in the Introduction. Specifically, previous

work has found positive, negative, and no relationship between soil carbon and yields. Our work is not designed to resolve which of these patterns is correct because we believe that those site-specific relationships capture local realities. Rather, we are trying to capture global, average relationships that can help quantify the relationship between SOC and yield for broad-scale policy targets.

L. 141. Authors argue that two thirds of maize and wheat cultivation takes place on soils with less than 2 % SOC. What is, for comparison, the average % SOC of croplands worldwide? Are these two staple crops planted on particularly C-poor soils?

As our study focuses on two of the most important staple crops that are planted globally, we chose to focus on the SOC contents for maize and wheat. We can include the average SOC concentration on croplands worldwide in our supplemental file.

L. 160. Are the authors aware of any long-term field experiment where an increase from 0.5 to 2 % SOC has been observed? This seems unlikely to me. Even a doubling (previous sentence) is ambitious. The following argumentation, that higher SOM soils may supply enough plant available nutrients to sustain crop yields with drastically cutting back N fertilizer input overlooks that these are typical situations of SOM decline, as observed in many long-term experiments, where plant productivity can be maintained at low nutrient input rates only because of SOM decline and the associated release of organically bound nutrients.

This is a good point, and we recognize that building SOC from 0.5 to 2.0% represents a very large increase. Such an increase would require a significant amount of inputs that may not be feasible due to inherent and logistical difficulties related to soil properties, climate, and farmer access to inputs (lines 216-228). We will further stress the challenges associated with increasing SOC, while also highlighting experimental results recently published that show a range of annual increases in SOC for temperate agricultural soils (Poulton et al., 2018). The annual increases reported in this study range from 0.3 to 18% and are a result of a number of different inputs ranging from farmyard

manure to mineral fertilization, some of which the authors of this study acknowledge may not be practical for farmers.

L. 193 ff. The first para in section 2.3. belongs largely to the method section and is partially a repetition of that.

We will revise this text in a revision so as to avoid repetition with our Methods section.

L. 215. It is not clear where the yield gap comes from – how was it calculated, was it taken from the literature? Clarification needed.

We will provide more context and the relevant citation for our yield gap analysis. Specifically, we are using a global data set (Mueller et al., 2012) that provides a global assessment of the difference between observed yields and attainable yields.

L. 302. Authors refer to Söderström et al. 2014. I looked up that reference where I could not find a database as key repository but rather a research approach. Should be clarified.

Thank you for pointing this out. Söderström et al. is another manuscript from this database effort, but we will cite the original paper in a revised version of our manuscript (Haddaway et al., 2015).

L. 352. I suggest to use three classes: rainfed, irrigated, unknown.

When extracting data, in cases where authors did not specify how crops were watered, we scored them as rainfed. We will revisit the papers and revise our data set to reflect this uncertainty.

L. 353. Filling data gaps for soil pH and texture for experimental sites by a global database may introduce large errors and, potentially, biased estimates, given that these soil properties vary much over short distances. I suggest to either exclude those variables as explanatory ones or to ask authors of the studies to provide those data for their sites. Alternatively, these parameters can be categorized and used as categorical

variables.

We note that many of the studies were published prior to recent initiatives to deposit data products for published papers, making the kind of analysis we did additionally challenging. As such, we acknowledge that using values from a global database is not ideal and do acknowledge this as a limitation with our manuscript (lines 240-242). We did contact all authors for meta-data and raw data from their published studies, however, we only received data from three of the authors. As part of our original data exploration, we calculated the correlation coefficient for pH ($r = 0.83$) and soil texture ($r = 0.61$) between SoilGrids data and measured data from experimental studies in our data set. We also ran our regression model without texture and pH, and the coefficients on our model terms were essentially unchanged. We chose to retain these terms, however, because we believe that they do have established biological mechanisms as to their influence on yield. Furthermore, the range of both pH and percent clay data observed in SoilGrids reflects the range of data observed in our data set. Therefore, we believe that the relationships between variables are transferable between data sets even if the two data sets predict different values for the same place.

Table 2. I suggest to add a percentage increase in production from an increase in SOC to the table to make the global yield average and the increase in production comparable to each other.

This is a good suggestion and one we will include in a revised manuscript.

Figure 2. Not clear why the figure relates to maize and yield in line 1114 whereas the caption in line 1115 refers to maize only.

Thank you for catching this. We will revise the figure title to say, "Relationship between SOC and yield of maize for published studies."

Figure 5. The figure is interesting but results would better be presented as percentage increase in yield, and not as percentage closure of yield gap. The yield gap itself is

prone to large uncertainty, both in extent and possible reasons, and these uncertainties are not explicitly included.

When making our figures, we did create a map that featured percentage yield increase, however, it was difficult to visualize gains when presented at the broad global scale. We believe the yield gap map provides a clearer illustration of the areas that stand to gain the most in terms of identifying impacts of SOC on yield.

Figure 4. The provided interpretation of this results ignores the fact that building up additional SOC requires additional N.

This is a good point, and as we mention above, we will provide more discussion related to the challenges of building SOM/SOC, and that in may require the addition of inorganic N or organic N amendments.

References

Adhikari, K. and Hartemink, A. E.: Linking soils to ecosystem services - A global review, Geoderma, 262, 101–111, doi:10.1016/j.geoderma.2015.08.009, 2016.

Chabbi, A., Lehmann, J., Ciais, P., Loescher, H. W., Cotrufo, M. F., Don, A., San-Clements, M., Schipper, L., Six, J., Smith, P. and Rumpel, C.: Aligning agriculture and climate policy, Nature Clim. Change, 7(5), 307–309, 2017.

Giller, K. E., Witter, E., Corbeels, M. and Tittonell, P.: Conservation agriculture and smallholder farming in Africa: The heretics' view, Field Crop Res., 114(1), 23–34, 2009.

Haddaway, N. R., Hedlund, K., Jackson, L. E., Kätterer, T., Lugato, E., Thomsen, I. K., Jørgensen, H. B. and Söderström, B.: What are the effects of agricultural management on soil organic carbon in boreo-temperate systems? Environmental Evidence 2014 3:1, 4(1), 23, doi:10.1186/s13750-015-0049-0, 2015.

Hatfield, J. L., Sauer, T. J. and Cruse, R. M.: Soil: The Forgotten Piece of the Water, Food, Energy Nexus, Adv. Agron., 143, 1–46, doi:10.1016/bs.agron.2017.02.001,

2017.

Mueller, N. D., Gerber, J. S., Johnston, M., Ray, D. K., Ramankutty, N. and Foley, J. A.: Closing yield gaps through nutrient and water management, Nature, 490(7419), 254–257, doi:10.1038/nature11420, 2012.

NRC: Understanding agricultural sustainability, in Toward Sustainable Agricultural Systems in the 21st Century, pp. 1–29, National Academies Press, Washington, DC. 2010.

Palm, C. A., Giller, K. E., Mafongoya, P. L. and Swift, M. J.: Management of organic matter in the tropics: translating theory into practice, Nutr. Cycl. Agroecosys., 61(1997), 63–75, doi:10.1023/A:1013318210809, 2001.

Pittelkow, C. M., Liang, X., Linquist, B. A., van Groenigen, K. J., Lee, J., Lundy, M. E., van Gestel, N., Six, J., Venterea, R. T. and van Kessel, C.: Productivity limits and potentials of the principles of conservation agriculture, Nature, 517(7534), 365–368, doi:10.1038/nature13809, 2014.

Poulton, P., Johnston, J., Macdonald, A., White, R. and Powlson, D.: Major limitations to achieving "4 per 1000" increases in soil organic carbon stock in temperate regions: Evidence from long-term experiments at Rothamsted Research, United Kingdom, Glob. Change Biol., 24(6), 2563–2584, doi:10.1111/gcb.14066, 2018.

Trabucco, A., and Zomer, R.J. 2009. Global Aridity Index (Global-Aridity) and Global Potential Evapo-Transpiration (Global-PET) Geospatial Database. CGIAR Consortium for Spatial Information. Published online, available from the CGIAR-CSI GeoPortal at: http://www.csi.cgiar.org.

---

## Referee Comment (RC2) · Anonymous Referee #2 · 21 Sep 2018

This study examines the relationship between SOM and yields of wheat and maize across a range of agroecological contexts around the globe. The authors then apply this relationship to better understand the potential of increased SOM stocks to improve yields, as well as reduce N fertilizer inputs.

The study is ambitious in scope and their approach involved a number of assumptions and simplifications, and therefore requires considerable caution in the interpretation of their findings. Despite these drawbacks, I appreciated the effort and feel that the study represents a valuable and novel contribution towards addressing a complex issue with relevance to global agricultural sustainability. While I enjoyed this paper, I have several comments/critiques for the authors.

General comments:

The premise that increased SOM will reduce N inputs seems a bit misleading. Both the building of SOM (to 2% SOC) and its continued maintenance at this higher level will require considerable quantities of organic matter inputs both now and into the foreseeable future. So it seems unlikely that total N inputs will actually decrease, but really we are talking about a shift from inorganic to organic N sources. The authors allude to this in several places, but it could be spelled out more clearly. In reading the authors' responses to Reviewer 1, it seems that they now better recognize the need to address this.

Related to this, the study largely ignores the dynamic state of SOM. For example, soils in a state of rapid SOM decline may actually be supporting yields better than a soil at a similar level of SOM, simply because more nutrients are being mineralized as this SOM is lost.

I appreciate Fig. 1 showing origin of the datasets considered in this study, but am a little concerned about the high number of observations from China and how this might bias the findings. This should be addressed in the discussion.

Related to the above comment, it would be nice to see a table that provides a breakdown of how the sites were distributed in terms of number of sites with and without irrigation and with wheat vs. corn, as well as different ranges of pH, aridity, clay content, latitude, so that readers can better assess potential biases in the dataset on their own. This could be a new table in the main text or alternatively in the supplementary materials.

I understand the value of keeping the model relatively simple, but was surprised that several potentially important interaction terms were left out, while others (i.e., SOM x N input) where included. For example, I would expect to see a strong interaction between SOM and irrigation, such that SOM would be more important in rain-fed systems (particularly in semi-arid regions) than in irrigated agriculture, where the water related benefits of SOM would be less important. Also, I would have expected the different

crop types (and potentially sandy vs. clay textured soils) to respond differently to vary-ing SOM levels. Please consider including these terms or at least explain why the SOM x N interaction was included in the model and some of these other terms not.

Specific comments:

L112: the reported value of 0.25 is not very informative here in the text without providing units or some sort of additional explanation.

L116-121: the logic behind the sentence "the asymptotic relationship between SOC and yield lends support to the idea that building SOC will increase yields – at least to a certain extent – as opposed to simply being an outcome of higher yields." Is not entirely clear. Could it not be that yields have a larger effect on building SOM at higher levels? These two sentences should be perhaps omitted or further clarified.

L133: It seems the asymptotic relationship and leveling off above 2% (in Fig 2) may be strongly influenced by relatively few observations and I wonder if the authors conducted an sort of leverage tests (e.g., Cook's distance) to examine the potential influence of extreme observations. This is especially notable for the 4-5 sites that were at or above 2.5% SOC and with very low fertilizer addition and yields (in the bottom right corner of Fig. 2).

Also, It is not entirely clear how inter-annual variability was taken into account, espe-cially for rain-fed sites, where a severe drought in the year of yield data collection could drastically skew results.

L154: specify that your are referring more to 'inorganic' of 'synthetic N inputs'

L155: suggest replacing 'achieving' with 'obtaining', as crops to not achieve nutrients they obtain them.

L159-163: As mentioned above, the authors should acknowledge that higher SOC is not necessarily allowing for lower total N inputs, but perhaps lower synthetic N inputs, since there is likely to be relatively higher inputs of organic matter (and organic N) in

soils with higher SOC, or at least there should be if are managed in a way that that seeks to maintain these levels of SOC.

L164-165: Again, why where interactions only examined between SOC and N input and not for other factors that are very likely to interact with SOC, such as irrigation, crop type, texture, and aridity?

L165-167: Could this also have to do with Liebig's law of the minimum, such that higher SOM levels are really just supplying more P, K and other essential nutrients that may be co-limiting to N at higher N levels, but not at low N application levels. Please clarify

L188-191: What is this calculation and the 3.73 million tons based on? Please elaborate.

L233: yes, water retention is important, but also improved nutrient (especially N) supply from decaying SOM

L372-374: Again, based on this section and Table 1, what was rationale for including the SOM x N interaction? Also, as mentioned above why were other variables, that were likely to strongly interact with SOM (e.g., irrigation, clay, crop type), not included? This seems rather arbitrary and inclusion of these other interaction could have helped explain significant variability in yield across sites.

Fig 1: again need to the discuss the potential bias of having so many sites from one country, China, especially since fertilizer inputs in China are typically much higher than other parts of the world.

Fig 2: this is confusing, as the title suggest that the relationship includes maize and wheat, but then the next sentence says that its just for rain-fed maize. Please clarify.

Fig 4: which crops/conditions are being presented here. As for Fig 2, this needs to be better clarified.

Fig 5: the numbers on top of the colored boxes are small and difficult to read, especially

when printed in B&W

---

## Author Comment (AC2) · 1 Oct 2018

Response to Reviewer 2

This study examines the relationship between SOM and yields of wheat and maize across a range of agroecological contexts around the globe. The authors then apply this relationship to better understand the potential of increased SOM stocks to improve yields, as well as reduce N fertilizer inputs.

The study is ambitious in scope and their approach involved a number of assumptions and simplifications, and therefore requires considerable caution in the interpretation of their findings. Despite these drawbacks, I appreciated the effort and feel that the study represents a valuable and novel contribution towards addressing a complex issue with

relevance to global agricultural sustainability. While I enjoyed this paper, I have several comments/critiques for the authors.

RESPONSE: Thank you for this overall positive assessment. We do indeed view our work as a step forward in addressing a complex issue with relevance – academically, for policy, and for practice – to global agricultural sustainability. Equally, we appreciate the limitations of our work. We believe that our response to your comments below will ensure that we detail these limitations openly in our manuscript so that the advance we offer can be built upon constructively to evaluate the inferences we make.

General comments:

The premise that increased SOM will reduce N inputs seems a bit misleading. Both the building of SOM (to 2% SOC) and its continued maintenance at this higher level will require considerable quantities of organic matter inputs both now and into the foreseeable future. So it seems unlikely that total N inputs will actually decrease, but really we are talking about a shift from inorganic to organic N sources. The authors allude to this in several places, but it could be spelled out more clearly. In reading the authors' responses to Reviewer 1, it seems that they now better recognize the need to address this.

RESPONSE: Thank you for this comment. As Reviewer 2 mentions, this also came up with Reviewer 1. In our revision, we will address this more comprehensively. Specifically, as we mentioned in response to Reviewer 1, we recognize that a combination of both organic and inorganic nutrients will be necessary to both build SOM and improve crop yields. Building SOM and cutting back on N fertilizer will require that the SOM-N mineralization compensates for the reductions in mineral N fertilizer, and we will state this in our revised manuscript.

Related to this, the study largely ignores the dynamic state of SOM. For example, soils in a state of rapid SOM decline may actually be supporting yields better than a soil at a similar level of SOM, simply because more nutrients are being mineralized as this

SOM is lost.

RESPONSE: This is a good point, and we will provide further explanation in our discussion related to the "soil carbon dilemma" (Janzen, 2006). Namely, that to derive the benefits of SOM, it must be mineralized and used. As Janzen mentions in this paper, to both build SOM and derive nutrients from it, continuous inputs of organic matter will be needed to account for that which is lost through mineralization. We will ensure that our revised discussion highlights this expected effect of SOM (i.e. nutrient supply), but will also make sure to highlight that SOM is expected to have positive effects on productivity for other reasons (e.g. improved aeration and moisture supply). As such, a soil with rapid SOM decline could provide more nutrients but also could be limiting for other reasons. We will call for controlled experimental work to help uncouple such mechanisms.

I appreciate Fig. 1 showing origin of the datasets considered in this study, but am a little concerned about the high number of observations from China and how this might bias the findings. This should be addressed in the discussion.

RESPONSE: We do have a large proportion of studies from China in our data set as is highlighted in Fig. 1. We state, however, in our methods that the variation observed in our data set for our model parameters reflects that observed within the global data sets we used for our extrapolations (lines 443-447). However, we agree this is an important point when making "global generalities," and in a revised discussion we will emphasize the need for studies to inform such understanding to come more evenly from systems where wheat and maize are grown.

Related to the above comment, it would be nice to see a table that provides a breakdown of how the sites were distributed in terms of number of sites with and without irrigation and with wheat vs. corn, as well as different ranges of pH, aridity, clay content, latitude, so that readers can better assess potential biases in the dataset on their own. This could be a new table in the main text or alternatively in the supplementary

materials.

RESPONSE: This is a good idea, and we will include such a table in our supplemental materials. Furthermore, we have uploaded our entire data set to KBS repository and provided a link to it within our manuscript for anyone to view and use. This way, readers can explore the data to see the breakdown in variables as Reviewer 2 mentions.

I understand the value of keeping the model relatively simple, but was surprised that several potentially important interaction terms were left out, while others (i.e., SOM x N input) where included. For example, I would expect to see a strong interaction between SOM and irrigation, such that SOM would be more important in rain-fed systems (particularly in semi-arid regions) than in irrigated agriculture, where the water related benefits of SOM would be less important. Also, I would have expected the different soils with higher SOC, crop types (and potentially sandy vs. clay textured soils) to respond differently to varying SOM levels. Please consider including these terms or at least explain why the SOM x N interaction was included in the model and some of these other terms not.

RESPONSE: When we initially created our model, we did not include any interactions because of the sheer number of potential interactions that could be included, which would take up too many degrees of freedom. Additionally, and perhaps more importantly, 3-and-more way interactions are very hard to disentangle and their statistical significance should be interpreted with caution (Gelman and Loken, 2013). Following the philosophical and operational statistical methods we adopted ((Hobbs and Hilborn, 2006) – cited in our methods), we limited most of our exploration to only two-way interactions where we had a strong ecological rationale for expected effects. As such, we decided to include an SOM x N interaction to specifically explore potential reductions in N fertilizer with increased SOC concentrations. This was an effort to see if there is a level of SOC that can compensate for N input. We will further justify our decision for including this interaction and not others in a revised version of our manuscript. However, we do acknowledge that the interactions Reviewer 2 suggests present interesting lines

of inquiry. We re-ran our regression model with additional interactions to include SOC x irrigation, SOC x clay, and SOC x aridity. These interactions did not offer any additional explanatory power (the r2 was essentially unchanged and the coefficients were small). Furthermore, the main findings between SOC, N inputs, and yield were essentially unchanged with additional interactions. As our main findings remain the same with and without these additional interactions, we are choosing to maintain our analysis as it is. We will provide as supplementary material a table that shows the lack of sensitivity of SOC and N input effects to inclusion of these additional two-way interactions, justifying the regression model results we focus on.

Specific comments: L112: the reported value of 0.25 is not very informative here in the text without providing units or some sort of additional explanation.

RESPONSE: This number was meant to point out the fact that the slope of the relationship between SOC and yield levels off at 2% SOC. We will clarify this point in our revision.

L116-121: the logic behind the sentence "the asymptotic relationship between SOC and yield lends support to the idea that building SOC will increase yields – at least to a certain extent – as opposed to simply being an outcome of higher yields." Is not entirely clear. Could it not be that yields have a larger effect on building SOM at higher levels? These two sentences should be perhaps omitted or further clarified.

RESPONSE: This discussion was meant to highlight the challenges of quantifying the relationship between SOM and yield since the relationship could potentially be causative in both directions, with greater SOC leading to higher yields but also higher yields increasing SOC concentrations. This sentence was intended to demonstrate SOC as a cause – at least to some extent – of higher yields in the case of our analysis. For instance, if yield was on the x-axis as an explanatory variable for SOC on the y-axis, we would expect higher yields to keep driving higher levels of SOC (i.e. the relationship would appear more linear) since we know that soils can accumulate concentrations much greater than 2% (i.e. we are at a point that is well below theoretical and/or empirical soil C saturation points); however, our data do not display this pattern and higher yields do not appear to be driving higher levels of SOC. We will clarify this point in our revision.

L133: It seems the asymptotic relationship and leveling off above 2% (in Fig 2) may be strongly influenced by relatively few observations and I wonder if the authors conducted any sort of leverage tests (e.g., Cook's distance) to examine the potential influence of extreme observations. This is especially notable for the 4-5 sites that were at or above 2.5% SOC and with very low fertilizer addition and yields (in the bottom right corner of Fig. 2).

RESPONSE: We did not perform any leverage tests for our initial analysis. However, as suggested by Reviewer 2, we did evaluate Cook's distance and did not find any influential data points that would significantly change our regression relationship. Additionally, we re-ran our regression after removing the 4 data points in question, and model coefficients remained essentially the same. We will mention this in the revised methods and in the legend of Fig. 2.

Also, It is not entirely clear how inter-annual variability was taken into account, especially for rain-fed sites, where a severe drought in the year of yield data collection could drastically skew results.

RESPONSE: We nested year within site as a random effect in our regression model to account for spatial and temporal correlation. We will clarify this in our revision. There were some instances of low yields at rain-fed sites within our data set. We believe these observations are important to include as they capture the local realities of the relationship between climatic variables and yield. Our data set then uses these locally observed data points to capture a global average relationship between SOC and yield, which we state in the paper will need to be built on at sub-regional scales to provide data directly relevant to farmers and land managers.

L154: specify that your are referring more to 'inorganic' of 'synthetic N inputs'

RESPONSE: We will make this change in a revision.

L155: suggest replacing 'achieving' with 'obtaining', as crops to not achieve nutrients they obtain them.

RESPONSE: We will make this edit in a revision.

L159-163: As mentioned above, the authors should acknowledge that higher SOC is not necessarily allowing for lower total N inputs, but perhaps lower synthetic N inputs, since there is likely to be relatively higher inputs of organic matter (and organic N) in soils with higher SOC, or at least there should be if are managed in a way that that seeks to maintain these levels of SOC.

RESPONSE: We will make this point clear in a revision.

L164-165: Again, why where interactions only examined between SOC and N input and not for other factors that are very likely to interact with SOC, such as irrigation, crop type, texture, and aridity?

RESPONSE: As mentioned above, we chose to include only an SOC x N interaction since we were asking a question specifically related to the interaction between SOC and N as it relates to agricultural inputs. In the revised paper, we will include the additional regression analyses to show that including these interactions did not offer any more explanatory power to our model and our main findings (i.e., coefficient estimate sizes) essentially remained the same.

L165-167: Could this also have to do with Liebig's law of the minimum, such that higher SOM levels are really just supplying more P, K and other essential nutrients that may be co-limiting to N at higher N levels, but not at low N application levels. Please clarify

RESPONSE: This is a good point and one we will include as a way to explain why N had a greater impact on yield at higher SOC concentrations. Other benefits of higher

SOM include better moisture retention, improved structure and aeration, etc., so there is a substantive list of benefits expected for plants at higher SOM concentrations.

L188-191: What is this calculation and the 3.73 million tons based on? Please elaborate.

RESPONSE: We include the basis for this calculation in our methods (lines 495-502). In the revision, we will refer specifically to those lines in this section of the discussion.

L233: yes, water retention is important, but also improved nutrient (especially N) supply from decaying SOM

RESPONSE: This is a good point and one we will include in our revision.

L372-374: Again, based on this section and Table 1, what was rationale for including the SOM x N interaction? Also, as mentioned above why were other variables, that were likely to strongly interact with SOM (e.g., irrigation, clay, crop type), not included? This seems rather arbitrary and inclusion of these other interaction could have helped explain significant variability in yield across sites.

RESPONSE: We agree that there are a number of potential interactions between SOC and other variables included in our model. As mentioned above, we will now explain our decision and also include the additional regression output as a supplemental table.

Fig 1: again need to the discuss the potential bias of having so many sites from one country, China, especially since fertilizer inputs in China are typically much higher than other parts of the world.

RESPONSE: As mentioned above, in the revised discussion we will emphasize the need for studies to inform such understanding to come more evenly from systems where wheat and maize are grown. Further, we will highlight that our Chinese-site derived observations fall within the range of those we used for our global extrapolations. Therefore, we felt confident that potential bias for establishing a global, average relationship between SOC and yield was minimal in this instance.

Fig 2: this is confusing, as the title suggest that the relationship includes maize and wheat, but then the next sentence says that its just for rain-fed maize. Please clarify.

RESPONSE: Thank you for bringing this to our attention. Reviewer 1 also noticed this error, and we will edit our caption in a revision to specify that the regression relationship only includes rain-fed maize.

Fig 4: which crops/conditions are being presented here. As for Fig 2, this needs to be better clarified.

RESPONSE: We will clarify this in a revision as well to specify that the regression relationships are plotted for rain-fed maize.

Fig 5: the numbers on top of the colored boxes are small and difficult to read, especially when printed in B&W

RESPONSE: We will edit these to increase their font size for readability in our revision.

References Gelman, A. and Loken, E.: The garden of forking paths: Why multiple comparisons can be a problem, even when there is no "fishing expedition" or "p-hacking" and the research hypothesis was posited ahead of time. Retrieved from http://www.stat.columbia.edu/~gelman/research/unpublished/ p_hacking.pdf

Hobbs, N. T. and Hilborn, R.: Alternatives to statistical hypothesis testing in ecology: A guide to self teaching, Ecol. App., 16(1), 5–19.

Janzen, H.: The soil carbon dilemma: Shall we hoard it or use it? Soil Biol. Biochem., 38(3), 419–424, doi:10.1016/j.soilbio.2005.10.008, 2006.

---

## Author Response (AR1)

November 13, 2018

Dear Dr. Rumpel,

Thank you for the opportunity to submit a revised version of "Global meta-analysis of the relationship between soil organic matter and crop yields." We appreciate your handling of this manuscript and the constructive comments from the reviewers. We have revised the paper in response to your comment and those from the reviewers and have included our responses below. Please let us know if you have further questions or seek additional clarification on any points.

Sincerely,

Emily Oldfield (on behalf of all co-authors)
Yale School of Forestry and Environmental Studies
emily.oldfield@yale.edu

**Editor Comments to the Author:**

*The authors provided convincing answer to the reviewer comments except one point. The study is strongly biased towards contributions from China, where input of inorganic fertilisers can be very high. It could be good to run the model without studies from China to see if the relationships found are still valid.*

As suggested, we ran our model both excluding data points from China and also with only observations from China. We found that the effect size of N did not change significantly, however, the effect size of SOC did change depending on the subset of data analyzed. We found a stronger impact of SOC on yield when analyzing Chinese-only observations. The reason for this difference is that the majority (all but 10) of Chinese observations had SOC contents of less than 2%. Therefore, the strong effect size is capturing the range of SOC that appears to lead to the largest gains in yield. We now address these geographic differences in the discussion (lines 257-272) and also include regression output and figures within the supplementary material of our revision (specifically Table S2 and Figure S3). We believe this additional analysis highlights both the need for studies to come evenly from systems where maize and wheat are grown and also the importance of analyzing regional datasets that capture the observed range of SOC values in order to quantify a regionally-specific relationship between SOC and yield. We now emphasize these points within our discussion (lines 270-273).

**Response to reviewer comments (reviewer comments in italic)**

*The authors use a global data set on maize and wheat yields together with soil and other environmental variables to derive statistical relationships between SOC and yield.*
*The overall value of the study is appreciated. The interpretation of the data and observed*

*relationships is, however, going too far because direct evidence for the postulated effects, as it could be derived from long-term experiments at different SOC levels, cannot be derived and many other influencing factors were ignored.*

*Title and abstract. In both, SOM is described as the key variable but the study relies on SOC data. This should be reflected in the title and the abstract. This already touches a more fundamental problem – the study does not provide mechanistic insight as to why higher SOC results in higher yields. More SOC is often obtained using more organic inputs, i.e., more macro- and micro-nutrients bound to SOM. A second issue here, related to the first one is that, correctly, a higher SOC concentration might reduce the amount of N needed as fertilizer to get the same yield, but it is not discussed how much more N must be fertilized to reach the higher SOC level.*

We appreciate the reviewer's comments and have made revisions to address these concerns. The first major concern is that we do not provide mechanistic insight as to why SOM (or SOC) would increase yield. We believe that the mechanisms between SOM/SOC and crop yield have been well established, but poorly quantified. For instance, we would expect SOC to be associated with greater cation exchange capacity for the exchange of micronutrients and greater water holding capacity. SOC, because it is the majority constituent of SOM, is also highly correlated with macro-elements contained in SOM. The contribution of our project is not to tease apart the relative importance of the separate mechanisms by which SOM/SOC operates, though we do believe this would be a very important, but challenging, project. Instead, our aim is to establish relationships at broad scales between SOC and yield to provide better quantification of this relationship for policy initiatives such as the recently launched Global Soil Health Challenge. This has been identified as a critical knowledge gap among producers, policy makers, and researchers alike (Adhikari and Hartemink, 2016; Chabbi et al., 2017; Hatfield et al., 2017). For instance, the U.S. National Research Council stated in their 2010 report on sustainable agriculture that "measures of [SOM] are a cornerstone of most sustainability and soil quality assessments...However, the numerical level that would be considered good, or what change in [SOM] levels constitutes a significant functional change, has not been established (NRC, 2010)." Our paper is an attempt to answer that call, and we have made this clear in our revision (lines 90-91).

Secondly, the reviewer raises concerns related to the challenge of using observational differences across broad spatial scales to test SOC-yield relationships. We note here and now make clear in our revision (lines 98-100, 404-408) that the differences in soil carbon observed in our data set are from experimental plots capturing long-term differences in SOC within a given site. Specifically, our data capture differences within SOC in experimental plots largely driven by management interventions related to inputs (e.g. compost, fertilizer, manure, crop residues) and tillage (e.g. no-till versus till). We capture these site-specific differences in management with site-level random intercept terms.

Regarding our analysis of potential fertilizer reductions: we recognize that a combination of both organic and inorganic nutrients will be necessary to help build SOM and improve crop yields. We highlight this further in our revision and stress that building SOM and cutting back on N fertilizer will require achieving an agricultural N balance where SOM-N mineralization accounts for the reductions in mineral fertilizer (lines 192-201). This will depend on the amount and C:N ratios of inputs used in specific agricultural systems. We also note and provide relevant citations that this may prove especially challenging in smallholder systems where there is often lack of access to and insufficient quality of organic inputs (Giller et al., 2009; Palm et al., 2001).

*L. 96 and methods. It is not clear why authors only used aridity and latitude as variables related to climate. Yields are strongly related to rainfall and temperature, which are easily available variables.*

We chose to use aridity since it is a variable that is expressed as a function of precipitation, temperature, and potential evapo-transpiration. We now include this information and relevant citation in a revised manuscript (Trabucco 2009) (lines 396-398). We did initially include rainfall and temperature variables in our statistical model, but since they were highly correlated, we chose to leave them out and include aridity since it is derived from temperature and precipitation data. The use of aridity has been used in other large-scale yield studies (Pittelkow et al., 2014).

*L. 121. More recent literature suggests that higher yield is not coming along with higher plant residue inputs (e.g., Hirte et al. 2018 Agriculture Ecosystems Environment 265).*

This is a good point, and one we highlight in the Introduction. Specifically, previous work has found positive, negative, and no relationship between soil carbon and yields. Our work is not designed to resolve which of these patterns is correct because we believe that those site-specific relationships capture local realities. Rather, we are trying to capture global, average relationships that can help quantify the relationship between SOC and yield for broad-scale policy targets.

*L. 141. Authors argue that two thirds of maize and wheat cultivation takes place on soils with less than 2 % SOC. What is, for comparison, the average % SOC of croplands worldwide? Are these two staple crops planted on particularly C-poor soils?*

As our study focuses on two of the most important staple crops that are planted globally, we chose to focus on the SOC contents for maize and wheat. This two-thirds percentage reflects the dataset we collected from published literature as well: Namely, the majority of soils from our dataset contained SOC concentrations equal to or less than 2%. For comparison, when we explored the average SOC contents for each agro-ecological zone (AEZ) for our analysis, we found that all AEZs aside from tropical humid, temperate humid, and boreal systems all had average SOC contents of 2% or less.

*L. 160. Are the authors aware of any long-term field experiment where an increase from 0.5 to 2 % SOC has been observed? This seems unlikely to me. Even a doubling (previous sentence) is ambitious. The following argumentation, that higher SOM soils may supply enough plant available nutrients to sustain crop yields with drastically cutting back N fertilizer input overlooks that these are typical situations of SOM decline, as observed in*

*many long-term experiments, where plant productivity can be maintained at low nutrient input rates only because of SOM decline and the associated release of organically bound nutrients.*

This is a good point, and we recognize that building SOC from 0.5 to 2.0% represents a very large increase. Such an increase would require a significant amount of inputs that may not be feasible due to inherent and logistical difficulties related to soil properties, climate, and farmer access to inputs (lines 231-242). We now further stress the challenges associated with increasing SOC, while also highlighting experimental results recently published that show a range of annual increases in SOC for temperate agricultural soils (lines 163-174). The annual increases reported in this study range from 0.3 to 18% and are a result of a number of different inputs ranging from farmyard manure to mineral fertilization, some of which the authors of this study acknowledge may not be practical for farmers (Poulton et al., 2018).

*L. 193 ff. The first para in section 2.3. belongs largely to the method section and is partially a repetition of that.*

We have revised this text in our revision to avoid repetition with our Methods section.

*L. 215. It is not clear where the yield gap comes from – how was it calculated, was it taken from the literature? Clarification needed.*

We now provide more context and the relevant citation for our yield gap analysis. Specifically, we are using a global data set (Mueller et al., 2012) that provides a global assessment of the difference between observed yields and attainable yields (lines 220-222).

*L. 302. Authors refer to Söderström et al. 2014. I looked up that reference where I could not find a database as key repository but rather a research approach. Should be clarified.*

Thank you for pointing this out. Söderström et al. is another manuscript from this database effort, but we now cite the original paper in our revision (Haddaway et al., 2015).

*L. 352. I suggest to use three classes: rainfed, irrigated, unknown.*

When extracting data, in cases where authors did not specify how crops were watered, we scored them as rainfed. In light of this comment, we went back to each paper that we classified as rainfed within our database and found a number of data points (n=46) that did not explicitly provide any information regarding irrigation and/or rainfall patterns over the growing season. For these instances, we scored them as unknown and ran our regression models again. The coefficients did not significantly change and so we decided to leave irrigation data as is in order to provide the greatest number of observations for our analysis.

*L. 353. Filling data gaps for soil pH and texture for experimental sites by a global database may introduce large errors and, potentially, biased estimates, given that these soil properties vary much over short distances. I suggest to either exclude those variables as explanatory ones or to ask authors of the studies to provide those data for their sites. Alternatively, these parameters can be categorized and used as categorical variables.*

We note that many of the studies were published prior to recent initiatives to deposit data products for published papers, making the kind of analysis we did additionally challenging. As such, we acknowledge that using values from a global database is not ideal and do acknowledge this as a limitation with our manuscript (lines 273-279). We did contact all authors for meta-data and raw data from their published studies, however, we only received data from three of the authors. As part of our original data exploration, we calculated the correlation coefficient for pH ($r = 0.83$) and soil texture ($r = 0.61$) between SoilGrids data and measured data from experimental studies in our data set. We also ran our regression model without texture and pH, and the coefficients on our model terms were essentially unchanged. We chose to retain these terms, however, because we believe that they do have established biological mechanisms as to their influence on yield. Furthermore, the range of both pH and percent clay data observed in SoilGrids reflects the range of data observed in our data set. Therefore, we believe that the relationships between variables are transferable between data sets even if the two data sets predict different values for the same place.

*Table 2. I suggest to add a percentage increase in production from an increase in SOC to the table to make the global yield average and the increase in production comparable to each other.*

This is a good suggestion and one we now include in our revision.

*Figure 2. Not clear why the figure relates to maize and yield in line 1114 whereas the caption in line 1115 refers to maize only.*

Thank you for catching this. We have revised the figure title to say, "Relationship between SOC and yield of maize for published studies."

*Figure 5. The figure is interesting but results would better be presented as percentage increase in yield, and not as percentage closure of yield gap. The yield gap itself is prone to large uncertainty, both in extent and possible reasons, and these uncertainties are not explicitly included.*

When making our figures, we did create a map that featured percentage yield increase, however, it was difficult to visualize gains when presented at the broad global scale. We believe the yield gap map provides a clearer illustration of the areas that stand to gain the most in terms of identifying impacts of SOC on yield.

*Figure 4. The provided interpretation of this results ignores the fact that building up additional SOC requires additional N.*

This is a good point, and as we mention above, we now provide more discussion related to the challenges of building SOM/SOC, and that it may require the addition of inorganic N or organic N amendments (lines 192-201).

**Response to Reviewer 2**

*This study examines the relationship between SOM and yields of wheat and maize across a range of agroecological contexts around the globe. The authors then apply this relationship to better understand the potential of increased SOM stocks to improve yields, as well as reduce N fertilizer inputs.*

*The study is ambitious in scope and their approach involved a number of assumptions and simplifications, and therefore requires considerable caution in the interpretation of their findings. Despite these drawbacks, I appreciated the effort and feel that the study represents a valuable and novel contribution towards addressing a complex issue with relevance to global agricultural sustainability. While I enjoyed this paper, I have several comments/critiques for the authors.*

Thank you for this overall positive assessment. We do indeed view our work as a step forward in addressing a complex issue with relevance – academically, for policy, and for practice – to global agricultural sustainability. Equally, we appreciate the limitations of our work. We believe that our response to your comments below will ensure that we detail these limitations openly in our manuscript so that the advance we offer can be built upon constructively to evaluate the inferences we make.

*General comments:*

*The premise that increased SOM will reduce N inputs seems a bit misleading. Both the building of SOM (to 2% SOC) and its continued maintenance at this higher level will require considerable quantities of organic matter inputs both now and into the foreseeable future. So it seems unlikely that total N inputs will actually decrease, but really we are talking about a shift from inorganic to organic N sources. The authors allude to this in several places, but it could be spelled out more clearly. In reading the authors' responses to Reviewer 1, it seems that they now better recognize the need to address this.*

Thank you for this comment. As Reviewer 2 mentions, this also came up with Reviewer 1. In our revision, we address this more comprehensively. Specifically, as we mentioned in response to Reviewer 1, we recognize that a combination of both organic and inorganic nutrients will be necessary to both build SOM and improve crop yields. Building SOM and cutting back on N fertilizer will require that the SOM-N mineralization compensates for the reductions in mineral N fertilizer, and we now state this in our revised manuscript (lines 192-201).

*Related to this, the study largely ignores the dynamic state of SOM. For example, soils in a state of rapid SOM decline may actually be supporting yields better than a soil at a*

*similar level of SOM, simply because more nutrients are being mineralized as this SOM is lost.*

This is a good point, and we now provide further explanation in our discussion related to the "soil carbon dilemma" (Janzen, 2006). Namely, that to derive the nutrient benefits of SOM, it must be mineralized and used. As Janzen mentions in this paper, to both build SOM and derive nutrients from it, continuous inputs of organic matter will be needed to account for that which is lost through mineralization. Our revised discussion highlights this expected effect of SOM (i.e. nutrient supply), but also highlights that SOM is expected to have positive effects on productivity for other reasons (e.g. improved aeration and moisture supply). See lines 233-253.

*I appreciate Fig. 1 showing origin of the datasets considered in this study, but am a little concerned about the high number of observations from China and how this might bias the findings. This should be addressed in the discussion.*

We do have a large proportion of studies from China in our dataset as is highlighted in Fig. 1. We state, however, in our methods that the variation observed in our dataset for our model parameters reflects that observed within the global datasets we used for our extrapolations (lines 500-504). However, we agree this is an important point when making "global generalities," and in our revised discussion we emphasize the need for studies to inform such understanding to come more evenly from systems where wheat and maize are grown (lines 268-272). We also removed data points from China and performed our regression analysis on this amended data set. We found that the impact of N input on yield did not vary dramatically from that of our original model, however, the effect size of SOC on yield was smaller. We also performed our regression analysis on observations from China only. With this analysis, we found a stronger impact of SOC on yield than that for the entire dataset. The reason for this difference is that the majority (all but 10) of Chinese observations had SOC contents of less than 2%. Therefore, the strong effect size is capturing the range of SOC that appears to lead to the largest gains in yield. We now address these geographic differences in the discussion (lines 257-268) and also include regression output and figures from these additional analyses within the supplementary material of our revision (specifically Table S2 and Figure S3).

*Related to the above comment, it would be nice to see a table that provides a breakdown of how the sites were distributed in terms of number of sites with and without irrigation and with wheat vs. corn, as well as different ranges of pH, aridity, clay content, latitude, so that readers can better assess potential biases in the dataset on their own. This could be a new table in the main text or alternatively in the supplementary materials.*

This is a good idea, and we now include such a table in our supplemental materials. Furthermore, we have uploaded our entire dataset to KNB repository and provided a link to it within our manuscript for anyone to view and use. This way, readers can explore the data to see the breakdown in variables as Reviewer 2 mentions.

*I understand the value of keeping the model relatively simple, but was surprised that*

*several potentially important interaction terms were left out, while others (i.e., SOM x N input) where included. For example, I would expect to see a strong interaction between SOM and irrigation, such that SOM would be more important in rain-fed systems (particularly in semi-arid regions) than in irrigated agriculture, where the water related benefits of SOM would be less important. Also, I would have expected the different soils with higher SOC, crop types (and potentially sandy vs. clay textured soils) to respond differently to varying SOM levels. Please consider including these terms or at least explain why the SOM x N interaction was included in the model and some of these other terms not.*

When we initially created our model, we did not include any interactions because of the sheer number of potential interactions that could be included, which would take up too many degrees of freedom. Additionally, and perhaps more importantly, 3-and-more way interactions are very hard to disentangle and their statistical significance should be interpreted with caution (Gelman and Loken, 2013). Following the philosophical and operational statistical methods we adopted (Hobbs and Hilborn, 2006) – cited in our methods – we limited most of our exploration to only two-way interactions where we had a strong ecological rationale for expected effects. As such, we decided to include an SOM x N interaction to specifically explore potential reductions in N fertilizer with increased SOC concentrations. This was an effort to see if there is a level of SOC that can compensate for N input. We now further justify our decision for including this interaction and not others in our revision (lines 436-445). However, we do acknowledge that the interactions Reviewer 2 suggests present interesting lines of inquiry. We re-ran our regression model with additional interactions to include SOC x irrigation, SOC x clay, and SOC x aridity. These interactions did not offer any additional explanatory power (the $r^2$ was essentially unchanged and the coefficients were small). Furthermore, the main findings between SOC, N inputs, and yield were essentially unchanged with additional interactions. As our main findings remain the same with and without these additional interactions, we are choosing to maintain our analysis as it is. We now provide as supplementary material a table (Table S3) that shows the lack of sensitivity of SOC and N input effects to inclusion of these additional two-way interactions, justifying the regression model results we focus on.

*Specific comments:*
*L112: the reported value of 0.25 is not very informative here in the text without providing units or some sort of additional explanation.*

This number was meant to point out the fact that the slope of the relationship between SOC and yield levels off at 2% SOC.

*L116-121: the logic behind the sentence "the asymptotic relationship between SOC and yield lends support to the idea that building SOC will increase yields – at least to a certain extent – as opposed to simply being an outcome of higher yields." Is not entirely clear. Could it not be that yields have a larger effect on building SOM at higher levels? These two sentences should be perhaps omitted or further clarified.*

This discussion was meant to highlight the challenges of quantifying the relationship between SOM and yield since the relationship could potentially be causative in both directions, with greater SOC leading to higher yields but also higher yields increasing SOC concentrations. This sentence was intended to demonstrate SOC as a cause – at least to some extent – of higher yields in the case of our analysis. For instance, if yield was on the x-axis as an explanatory variable for SOC on the y-axis, we would expect higher yields to keep driving higher levels of SOC (i.e. the relationship would appear more linear) since we know that soils can accumulate concentrations much greater than 2% (i.e. we are at a point that is well below theoretical and/or empirical soil C saturation points); however, our data do not display this pattern and higher yields do not appear to be driving higher levels of SOC. We now clarify this point in our revision (lines 117-122).

*L133: It seems the asymptotic relationship and leveling off above 2% (in Fig 2) may be strongly influenced by relatively few observations and I wonder if the authors conducted any sort of leverage tests (e.g., Cook's distance) to examine the potential influence of extreme observations. This is especially notable for the 4-5 sites that were at or above 2.5% SOC and with very low fertilizer addition and yields (in the bottom right corner of Fig. 2).*

We did not perform any leverage tests for our initial analysis. However, as suggested by Reviewer 2, we did evaluate Cook's distance and did not find any influential data points that would significantly change our regression relationship. Additionally, we re-ran our regression after removing the 4 data points in question, and model coefficients remained essentially the same. We now mention this in the revised methods (lines 422-424).

*Also, It is not entirely clear how inter-annual variability was taken into account, especially for rain-fed sites, where a severe drought in the year of yield data collection could drastically skew results.*

We nested year within site as a random effect in our regression model to account for spatial and temporal correlation (lines 408-410). There were some instances of low yields at rain-fed sites within our data set. We believe these observations are important to include as they capture the local realities of the relationship between climatic variables and yield. Our data set then uses these locally observed data points to capture a global average relationship between SOC and yield, which we state in the paper will need to be built on at sub-regional scales to provide data directly relevant to farmers and land managers.

*L154: specify that your are referring more to 'inorganic' of 'synthetic N inputs'*

We have made this change.

*L155: suggest replacing 'achieving' with 'obtaining', as crops to not achieve nutrients they obtain them.*

We have edited this sentence.

*L159-163: As mentioned above, the authors should acknowledge that higher SOC is not necessarily allowing for lower total N inputs, but perhaps lower synthetic N inputs, since there is likely to be relatively higher inputs of organic matter (and organic N) in soils with higher SOC, or at least there should be if are managed in a way that that seeks to maintain these levels of SOC.*

We have now made this point clear in our revision.

*L164-165: Again, why where interactions only examined between SOC and N input and not for other factors that are very likely to interact with SOC, such as irrigation, crop type, texture, and aridity?*

As mentioned above, we chose to include only an SOC x N interaction since we were asking a question specifically related to the interaction between SOC and N as it relates to agricultural inputs. In our revision's supplementary materials, we now include the additional regression analyses to show that including these interactions did not offer any more explanatory power to our model and our main findings (i.e. coefficient estimate sizes) essentially remained the same.

*L165-167: Could this also have to do with Liebig's law of the minimum, such that higher SOM levels are really just supplying more P, K and other essential nutrients that may be co-limiting to N at higher N levels, but not at low N application levels. Please clarify*

This is a good point and one we now include as a way to explain why N had a greater impact on yield at higher SOC concentrations (lines 180-182). Other benefits of higher SOM include better moisture retention, improved structure and aeration, etc., so there is a substantive list of benefits expected for plants at higher SOM concentrations.

*L188-191: What is this calculation and the 3.73 million tons based on? Please elaborate.*

We include the basis for this calculation in our methods (lines 552-559). In the revision, we will refer specifically to the Methods in this section of the discussion.

*L233: yes, water retention is important, but also improved nutrient (especially N) supply from decaying SOM*

This is a good point and one we now include in our revision (lines 252).

*L372-374: Again, based on this section and Table 1, what was rationale for including the SOM x N interaction? Also, as mentioned above why were other variables, that were likely to strongly interact with SOM (e.g., irrigation, clay, crop type), not included? This seems rather arbitrary and inclusion of these other interaction could have helped explain significant variability in yield across sites.*

We agree that there are a number of potential interactions between SOC and other variables included in our model. As mentioned above, we now explain our decision and also include the additional regression output as a supplemental table (Table S3).

*Fig 1: again need to the discuss the potential bias of having so many sites from one country, China, especially since fertilizer inputs in China are typically much higher than other parts of the world.*

As mentioned above, in the revised discussion we emphasize the need for studies to inform such understanding to come more evenly from systems where wheat and maize are grown. We have also provided additional discussion and supplemental analysis of our data when all observations from China were removed. We found that the impact of N is essentially the same for both regression analyses, however, the effect size for SOC is smaller for the amended data set (see Table S2). We now include more discussion addressing these points (lines 257-268).

*Fig 2: this is confusing, as the title suggest that the relationship includes maize and wheat, but then the next sentence says that its just for rain-fed maize. Please clarify.*

Thank you for bringing this to our attention. Reviewer 1 also noticed this error, and we have edited our caption to specify that the regression relationship only includes rain-fed maize.

*Fig 4: which crops/conditions are being presented here. As for Fig 2, this needs to be better clarified.*

We have clarified this to specify that the regression relationships are plotted for rain-fed maize.

*Fig 5: the numbers on top of the colored boxes are small and difficult to read, especially when printed in B&W*

We have edited these to increase their font size for readability.

References
Gelman, A. and Loken, E.: *The garden of forking paths: Why multiple comparisons can be a problem, even when there is no "fishing expedition" or "p-hacking" and the research hypothesis was posited ahead of time.* Retrieved from http://www.stat.columbia.edu/~gelman/research/unpublished/ p_hacking.pdf

[revised manuscript text omitted]